# Individual and population-level risk factors for new HIV infections among adults in Eastern and Southern Africa

Emma Slaymaker [1] ✉, Clara Calvert [2], Milly Marston[1], Kathryn Risher [3,4], Jeffrey W. Imai-Eaton [4,5], Louisa Moorhouse[4,6], Alison Price[1,7], Ramadhani Abdul[8], Albert Dube[7], Dorean Nabukalu[9], David Obor[10], Estelle McLean[7], Malebogo Tlhajoane[11], Keith Tomlin[1], Mark Urassa[12,24], Kathy Baisley[13,14], Amelia Crampin[1,7,15], Eveline Geubbels[8], Simon Gregson [4,6], Kobus Herbst[14,16], Daniel Kwaro[17], Tom Lutalo[9], Rob Newton [18], Jim Todd[19] & the ALPHA Network**

Despite substantial recent declines, general population HIV incidence in sub-Saharan Africa remains above international targets. Better description of risk factors for new infections would improve prioritisation of interventions. Using data from population-based cohorts in Kenya, Malawi, Tanzania, South Africa, Uganda, Zimbabwe we described the prevalence of risk factors for men and women aged 15-24 and 25-49 and estimated the association between individual and community-level risk factors and HIV acquisition between 2005 and 2016. Among 43,434 men and 55,919 women aged 15 to 49 there were 4,612 seroconversions. Education, marital status, male circumcision, new sexual partners, types of partner, prevalence of untreated HIV infection in the community and community partner acquisition rates were associated with HIV incidence. Only the prevalence of untreated HIV was a risk for both sexes and apparent at all ages. The prevalence of risk factors varied by age, sex and study. HIV incidence was higher in people aged 25-49 living in communities where men had high partner acquisition rates. Our results show potential for improved prevention through changed timing of prevention interventions relative to behaviour and the utility of using community characteristics to target prevention.

Despite dramatic declines in HIV incidence in Eastern and Southern Africa there were 450,000 new infections in 2023[1], the majority of which occurred among members of the general population[2]. UNAIDS 2030 targets for ending AIDS may be missed and HIV prevention programmes require strengthening[1]. Accurate targeting of prevention requires good information on who is most at risk of acquiring HIV and would help achieve these targets while being cost-effective to deliver.

Studies have explored risk factors for HIV acquisition in the general population across sub-Saharan Africa[3-27]. Generally, these found higher incidence in women compared to men and variations in incidence by age[3,11,14,22,27]. The most commonly identified risk factors for HIV infection were: being unmarried (in five populations)[6,10,11,15,18,20-22,26,28], having multiple partners (in four populations)[11,14,18,20-22,26,28,29], having a sexually transmitted infection (STI) (in six populations)[6,11,12,14,15,19,20,23,28,30-32] and alcohol use (in two populations)[6,11,15,20,21,26,33] (see Supplementary Material Appendix 1 for detailed description of literature review). However, comparison of risk factors across different populations is hampered by differences in

A full list of affiliations appears at the end of the paper. *A list of authors and their affiliations appears at the end of the paper.
✉e-mail: emma.slaymaker@lshtm.ac.uk

analytical approaches between the studies, in risk factors evaluated, period of study and the age ranges.

Understanding drivers of new infections is essential for effective prevention. Structural determinants and more proximate individual-level factors are relevant and some will be amenable to change. To identify potential risk factors for this analysis we devised an analytical framework (Supplementary Fig. 1), initially based on the proximate determinants framework set out by Boerma and Weir[34] adapted for an individual-based risk factor analysis and supported by our literature review. We start with the sequence of events needed for a susceptible person to become infected with HIV followed by a representation of how both the proximate and more distal risk factors influence different stages in that sequence. A risk factor may exert a differing degree of influence on different stages which might partly explain the varying results obtained in different analyses.

The sequence begins with a susceptible index person seeking, then meeting, a potential new sexual partner who may have HIV and may be viraemic. If they have sex, and do not use a method of HIV prevention, the susceptible person may become infected. If this does not happen, and the person remains susceptible, they may continue the partnership and the cycle will repeat. During the partnership the partner may become infectious for HIV and place the index person at risk. Within our framework we have identified the distal and proximate, measurable factors and which events in the sequence they may influence.

The prevalence of infectious individuals in the population and the chances of a susceptible person encountering an infectious one will greatly modify the effects of risk behaviours, such as sex without using a condom. The prevalence of infectious individuals depends on other people's risk behaviour, treatment uptake and adherence.

We investigated individual and community-level risk factors for incident HIV infections among adults in the general population in Kenya, Malawi, South Africa, Tanzania, Uganda and Zimbabwe. Kenya, Malawi, Tanzania and Zimbabwe are on track to achieve the UNAIDS 2030 target for reductions in incidence whilst South Africa has made moderate, and Uganda only limited progress towards this goal[35]. To understand the contribution of distinct, and ideally modifiable, behaviours and characteristics, we created variables that aligned with our analytical framework and were independent of other aspects of behaviour and individual circumstances (see Table 1). We looked for associations which were generalisable across settings in both the direction and magnitude of effect.

In this study, we find that education, marital status, male circumcision, new sexual partners, types of partner, prevalence of untreated HIV infection in the community and community partner acquisition rates are associated with HIV incidence. We show that the prevalence of risk factors and the nature of their associations with HIV incidence differs by age and sex.

## Results
### Data available
Between 2005 and 2016, 99,353 initially HIV-negative individuals aged 15 to 49 were observed and tested for HIV at least once following an initial negative HIV test (43,434 men and 55,919 women). They contributed 349,201 person years (58% from women) and there were 4,612 seroconversions (72% among women). The largest study, was uMkhanyakude (23,794 people and 2,876 serconversions) and the smallest Ifakara (1,451 people) (Table 2).

Data on sexual partners and recent behaviour was available for almost 90,000 person years. Coverage was incomplete, because data were not sought for all periods in all studies, and around 90% of respondents contributed time to the unknown category for these variables. There was less person time without behaviour data for men (e.g., 65% for number of partners) than for women (72%).

## Prevalence of potential risk factors
The distributions of person-years, by age and sex, for each of the potential risk factors are shown in for the pooled cohort in Table 2 and separately for each study, by sex, in Supplementary Data 1 and 2. These are described in detail in Supplementary Methods.

Most respondents lived in rural areas, or in small urban or peri-urban areas within a predominantly rural setting. Around a quarter of people reported moving house during the period of observation. People with primary education contributed approximately 60% of person-time.

Men and women differed in their behaviour and this also varied by age.

Young men spent 60% of their time with no partner, 30% with one and 10% with multiple partners. 95% of their time was prior to marriage. Young men also spent the least time in age disparate partnerships: 73% of their time in partnerships was with partners aged within 5 years of their own age and less than 1% was spent with partners where the age gap was more than 10 years. New partnerships were most common among young men who spent 23% of their time in partnerships that were less than a year old (Table 2 and Supplementary Data 1).

Older men spent the most time in partnerships, with only 10% of time without a partner, and the most in multiple partnerships (27% of time). A quarter of their time was prior to first marriage or cohabitation. Of their time spent in partnerships, 46% was with a partner within 5 years of their own age and 21% was with partners where the age gap was more than 10 years. New partnerships accounted for 20% of older men's time in partnerships (Table 2 and Supplementary Data 1).

Young women spent 48% of their time without a partner and 51% was with one partner. 75% of time was spent unmarried and 22% with a married or cohabiting partner. Age differences between partners were not universal and for 54% of young women's time in partnerships, the age gap was less than 5 years. 11% of young women's time was in the first 12 months of a partnership (Table 2 and Supplementary Data 2).

Older women spent 15% of their time with no partners and 84% with one partner. Around 60% of time was spent married or cohabiting with a partner, except in uMkhanyakude where most of the time was spent unmarried. Althought much of the time spent in partnerships was with partners similar in age (43%), substantial portions were with partners who had a 5 to 10 year age gap (28%) and those where the age gap was more than 10 years (17%). Just 6% of older women's time was spent in the first year of a partnership (Table 2 and Supplementary Data 2).

Information on partnership type was not regularly collected in every study. Where available, the proportion of time spent in regular partnerships was similar by age and sex, ranging from 22% to 30% of person-time in the pooled data (Table 2 and Supplementary Data 1 and 2). Older men spent a similar amount of time with casual partners (21%) but women and younger men reported less (10-12%).

Male circumcision status was also inconsistently collected but, where known, the proportion of young men's person-time post-circumcision ranged from 6% in Masaka to 92% in Ifakara. In total, 14% of the time contributed by young men for this analysis was known to be post-circumcision, 45% of the time where status was known. For older men, 18% of time was post-circumcision, representing 63% of the time for which status was known.

Differences between the studies were observed in marital status, numbers of partners, types of partner, new partnerships, and condom use (see Supplementary Data 1 and 2). The variation observed across the studies was not predictable, in that these aspects of behaviour did not differ in the same way between studies.

## Associations with HIV acquisition in pooled data
Table 3 gives the incidence rates (per thousand person years) derived from multiple imputation and crude hazard ratios (HR).

**Table 1 | List of measures identified in the conceptual framework with their corresponding definition in the data available for analysis**

| Framework level | Concept | Variable | Notes |
|---|---|---|---|
| **DISTAL** | | | |
| Characteristics | Age | 5-year age groups: 15–19, 20–24, 25–29, 30–34, 35–39, 40–44, 45–49 | Included as a time-varying covariate |
| Characteristics | Sex | Male, Female | Included |
| Characteristics | Education | No formal education, Primary, Secondary, Tertiary, Don't know | Included as a time-varying covariate[1] |
| Characteristics | Wealth | - | Not available |
| Circumstances | Demographic structure | Ratios of age group sizes | Not used - if estimated for whole study would simply be equivalent to study-level fixed effect. |
| Circumstances | Place of residence | Rural, Urban, Peri-urban, <1 km from road, >1 km from road agricultural estate, commercial center, roadside trading, subsistence farming | Included as a time-varying covariate: used in study-specific crude analysis, not harmonised |
| Circumstances | Moved house | Moved house within the last 12 months (binary) | Included as a time-varying covariate |
| Circumstances | Social context | Alcohol use | Not available |
| Circumstances | Infrastructure | Social premises | Not available |
| Circumstances | Health services | - | Not available |
| Time | Calendar year | Calendar year grouped (2005-2008, 2009-2012, 2013-2016) | Included as a time-varying covariate |
| Availability of partners | Community partner acquisition | Partner acquisition rate among peers per 100 person years | Included as a time-varying covariate |
| Availability of partners | Community partner loss | Partner loss rate among peers per 100 person years | Included as a time-varying covariate |
| Availability of partners | Community partner acquisition | Partner acquisition rate among potential partners per 100 person years | Included as a time-varying covariate |
| Availability of partners | Community partner loss | Partner loss rate among potential partners per 100 person years | Included as a time-varying covariate |
| Personal preference | | | |
| **PROXIMATE** | | | |
| Number of partners Type of partners | Marital status | Never married, currently married, formerly married, been married but no information on current status, unknown* | Included as a time-varying covariate[2] |
| Number of partners | Number of partners in the last year | 0,1,2, 3 +, Unknown* | Included as a time-varying covariate[3] |
| Number of partners | More than one partner in the last year | No, Yes, Unknown* | Included as a time-varying covariate[3] |
| Type of partners | Had a causal partner in the last year | No, Yes, Unknown* | Included as a time-varying covariate[3] |
| Type of partners | Had a regular (non-cohabiting) partner in the last year | No, Yes, Unknown* | Included as a time-varying covariate[3] |
| Type of partners | Age(s) of partner(s) in the last year | More than 10 year age gap, 5-10 year age gap, less than 5 year age gap | Included as a time-varying covariate[3] |
| Prevalence of untreated infection | Prevalence of untreated infection | Percentage of people of the opposite sex in the age range of potential partners who are HIV+ and not on treatment | Included as a time-varying covariate[3] |
| Contraception | Desire to conceive | | Not enough comparable data across studies |
| Contraception | Contraceptive use at last sex | No, Yes, Unknown* | Not enough comparable data across studies |
| HIV prophylaxis | Condom use with all partners in the last year | Consistent (used on all occasions with all partners), inconsistent (used consistently with some partners but not others, or sometimes used with a partner), not used, no partners | Included as a time-varying covariate[3] |
| HIV prophylaxis | Male circumcision | Circumcised (medical or traditional) or not | Included as a time-varying covariate |
| HIV prophylaxis | PrEP use | | Not included: little data, and little availability in the study areas during this period |
| Sexual Practices | sexual practices | | Not enough harmonisable data. |
| **SEQUENCE OF EVENTS** | | | |
| Seek new partner | | | Not enough harmonisable data. |
| Meet new partner | New partner in the last year | | Included as a time-varying covariate |
| Partner is viraemic | Viral load of partners | Proxied by untreated prevalence | Included. Not possible to collect this information for all types of partner |
| Exposure | Coital frequency with all partners in the last year | Average weekly coital frequency with all partners across the year | Included as a time-varying covariate[3]. Collected in Ifakara, Kisesa, Kisumu in two rounds only and in Manicaland in three rounds. |

**Table 1 (continued) | List of measures identified in the conceptual framework with their corresponding definition in the data available for analysis**

| Framework level | Concept | Variable | Notes |
|---|---|---|---|
| Outcome | HIV seroconversion | | |
| Continue partnership | Ended a partnership in the last year | Had a partnership that ended during the year preceding a survey round | Included as a time-varying covariate. In many survey rounds the only data on end of partnerships was whether or not it was ongoing at the time of the survey |

* Unknown includes time when information was not elicited from respondents (questions not asked) and information missing due to item non-response or respondents missing a survey round.
[1]Status changed at the midpoint between two reports.
[2]Status changed at the time the new status was reported.
[3]Status applied to the entire year before the survey, based on the responses given to questions about behaviour in that period.

The adjusted HR from the preferred multivariate models are shown in Table 4 and Supplementary Data 7–10 present the comparison multivariate models. The different versions of the models gave broadly similar results. Models including the prevalence of untreated infection and calendar year did not show an effect of both; we opted to include untreated prevalence in our preferred model, since it is the more proximate determinant of infection. For each sex and age group, this was model 4 in Supplementary Data 7–10, which included a fixed effect for study, the prevalence of untreated infection and the other risk factors but did not include age and calendar year.

In the crude analysis, associations with calendar year, prevalence of untreated infection, having a regular partner and having a new partner were seen for men and women in both age groups. Incidence was lower in 2013-16 compared to 2005-08 with crude HR ranging from 0.60 (95% CI 0.46-0.79) in older women to 0.70 (95% CI 0.61-0.81) in younger women (Table 3). Having new, regular or casual partnerships compared to all other partnership configurations (including no partner), were risks for HIV acquisition. There was strong evidence that the community prevalence of untreated HIV infection was associated with higher incidence hazards in all groups and this was the only association that remained apparent for men and women in both age groups in the adjusted for analysis (Table 4). The effect was strongest for young men where a one percentage point increase in this prevalence was associated with a 7% increase in the hazards of HIV acquisition (Adj. HR 1.07, 95% CI 1.05-1.09). The association with the prevalence of untreated infection was not seen in the models which included age and calendar time (Supplementary data 7 to 10). It is correlated both with age and with calendar time and, when included in the same models, these variables more efficiently explain the variation in the outcome.

In the crude analysis, characteristics of partnerships were consistently associated with acquisition risk for men and women in both age groups but once adjusted for other factors these associations differed by sex and by age. In the crude, there was strong evidence for an association with regular partners in all groups with HR ranging from 1.75 to 3.17. For casual partners the strongest evidence was for young women (crude HR 1.57 95% CI 1.22-2.02) and older women and men had only weak evidence for a more modest association (crude HR range 1.27-1.42). Strong evidence for associations with new partnerships were seen in all groups with HR ranging from 1.5 for men to 2.5 for older women (Table 3).

**Young men**

For young men, having moved house in the last year was associated with increased risk of HIV acquisition (crude HR 1.68 95% CI 1.11-2.55) but this was not seen in the adjusted model after adjusting for partnership factors. In both crude and adjusted models there was weak evidence that secondary education was protective (HR 0.72 95% CI 0.49-1.06) (Table 4).

The strong protective effects of being unmarried (adj HR 0.38 95% CI 0.23-0.61) and being circumcised (adj HR 0.38 95% CI 0.21-0.70)

were the only individual-level factors for which strong evidence of association with HIV acquisition was seen in the crude and adjusted analysis.

In the crude analysis, young men with no partners experienced a lower hazard of HIV acquisition than men with one partner (crude HR 0.62 95% CI 0.43-0.91) while those who had multiple partners (crude HR 2.32 95% CI 1.46-3.70) had higher risk. There was no evidence for an association with the age gap between partners. After adjustment, there was no evidence for the association of partner type with acquisition of HIV by young men with the adjusted HRs tending towards 1. There was weak evidence for a diminished association with multiple partners (Supplementary Data 9).

**Older men**

For older men, in the crude analysis there was weak evidence for increased risk of HIV acquisition following a house move (crude HR 1.30 95% CI 0.94-1.81) but, after adjustment, this could have been due to chance. A lower hazard of HIV acquisition among men with primary education in the crude analysis reversed, following adjustment, to a declining trend with increasing education but these associations may have been due to chance.

In the crude analysis, current marriage was protective for older men, with increased risks for the never and formerly married men but, once adjusted for having a partner, only the formerly married had higher risk of HIV acquisition (adj HR 1.41 95% CI 1.06-1.86). The HR for having multiple partners were positively associated with HIV acquisition but ths may have been due to chance (Supplementary Data 10). HIV acquisition hazard decreased as the age gap with partners increased, but this may also have been a chance finding (Table 3). Among older men, inconsistent condom use was associated with higher hazard of HIV acquisition in the crude analysis (HR 1.91 95% CI 1.35-2.69) but not in the adjusted. Unlike for younger men, there was no evidence for an association with male circumcision.

In the adjusted model having a regular partnership remained associated with risk for older men (Adj. HR 1.82 95% CI 1.24-2.68, Table 4) but the effect of new partnerships was diminished and the confidence intervals were wide (Supplementary Data 10).

**Young women**

For young women, having moved house in the last year showed a crude association with increased risk of HIV acquisition (crude HR 1.29 95% CI 1.04-1.60) which disappeared once adjusted for other factors. Secondary education was protective when compared with primary (adj. HR 0.74 95% CI 0.59-0.92). Being formerly married was associated with increased risk of HIV acquisition compared with current marriage (adj. HR 2.71 95% CI 1.63-4.50). Young women in casual partnerships had higher HIV incidence compared to young women with other partnership types (including no partner) (adj. 1.82 95% CI 1.38-2.39). Regular partners were also a risk for young women (adj. HR 1.70 95% CI 1.39-2.08). The association with new partnerships remained for young women (adj. HR 1.35 95% CI 1.07-1.72) but the increase in risk with

**Table 2 | Description of study population and prevalence of potential risk factors**

| | Men | | | | | | | | Women | | | | | | | |
|---|---|---|---|---|---|---|---|---|---|---|---|---|---|---|---|---|
| | 15-24 | | | | 25-49 | | | | 15-24 | | | | 25-49 | | | |
| | N | PY | S | M | N | PY | S | M | N | PY | S | M | N | PY | S | M |
| **Total** | 28937 | 75613 | 541 | | 19042 | 71769 | 763 | | 31924 | 77223 | 1843 | | 30702 | 124535 | 1470 | |
| **Study name** | | | | | | | | | | | | | | | | |
| Karonga (Malawi) | 2886 | 5718 | 8 | | 2584 | 6168 | 26 | | 3290 | 6322 | 26 | | 3577 | 8737 | 41 | |
| Kisesa (Tanzania) | 2409 | 6642 | 18 | | 1723 | 7940 | 58 | | 2396 | 5411 | 38 | | 3057 | 14968 | 109 | |
| Manicaland (Zimbabwe) | 1599 | 4861 | 17 | | 1788 | 6454 | 88 | | 1725 | 4546 | 56 | | 3108 | 12433 | 138 | |
| Masaka (Uganda) | 3038 | 8649 | 12 | | 2061 | 9418 | 61 | | 3608 | 8915 | 45 | | 2748 | 13354 | 79 | |
| Rakai (Uganda) | 4519 | 11451 | 59 | | 4097 | 18683 | 157 | | 5240 | 11912 | 132 | | 5210 | 24901 | 203 | |
| uMkhanyakude (South Africa) | 8552 | 23505 | 390 | | 2952 | 10950 | 300 | | 9604 | 26832 | 1469 | | 5670 | 24870 | 717 | |
| Kisumu (Kenya) | 5762 | 14452 | 36 | | 3580 | 11602 | 70 | | 5773 | 12772 | 70 | | 6547 | 23679 | 170 | |
| Ifakara (Tanzania) | 172 | 335 | 1 | | 257 | 554 | 3 | | 288 | 513 | 7 | | 785 | 1593 | 13 | |
| **Education** | | | | | | | | | | | | | | | | |
| No formal education | 121 | 245 | 3 | | 314 | 1071 | 13 | | 208 | 429 | 6 | | 1067 | 3639 | 36 | |
| Primary | 10807 | 23412 | 136 | | 6822 | 25496 | 222 | | 10821 | 20787 | 290 | | 11830 | 46926 | 359 | |
| Secondary | 6453 | 14006 | 61 | | 4522 | 13735 | 144 | | 7187 | 14830 | 182 | | 5021 | 17145 | 197 | |
| Tertiary | 238 | 358 | 4 | | 837 | 3054 | 25 | | 329 | 455 | 4 | | 750 | 2831 | 18 | |
| Unknown | 14975 | 37592 | 337 | | 7544 | 28414 | 357 | | 16580 | 40723 | 1361 | | 13498 | 53995 | 861 | |
| **Moved house in the last year** | | | | | | | | | | | | | | | | |
| No | 28502 | 70633 | 497 | | 18610 | 66960 | 698 | | 31279 | 70674 | 1688 | | 30081 | 117691 | 1336 | |
| Yes | 5798 | 4979 | 44 | | 5085 | 4811 | 65 | | 7787 | 6549 | 156 | | 7407 | 6845 | 135 | |
| **Calendar year (grouped)** | | | | | | | | | | | | | | | | |
| 2005-08 | 13008 | 23091 | 207 | | 10016 | 22111 | 285 | | 14401 | 24160 | 742 | | 15705 | 36775 | 472 | |
| 2009-12 | 17612 | 29980 | 207 | | 13679 | 29131 | 307 | | 18730 | 30093 | 635 | | 23010 | 50135 | 580 | |
| 2013-16 | 12708 | 22541 | 127 | | 9485 | 20527 | 170 | | 13675 | 22970 | 467 | | 16085 | 37626 | 419 | |
| **Partner acquisition rate among peers** | | 36748 | | 39.1 | | 51922 | | 24.1 | | 38579 | | 15.6 | | 72675 | | 6.3 |
| **Partner loss rate among peers** | | 36748 | | 22.1 | | 51922 | | 26.4 | | 38579 | | 5.6 | | 72675 | | 4.2 |
| **Partner acquisition rate in potential opposite sex partners** | | 34445 | | 16.5 | | 51922 | | 8.1 | | 38579 | | 42.0 | | 72432 | | 18.9 |
| **Partner loss rate in potential opposite sex partners** | | 34445 | | 5.9 | | 51922 | | 4.5 | | 38579 | | 27.9 | | 72341 | | 27.0 |
| **Current marital status** | | | | | | | | | | | | | | | | |
| Never married | 26246 | 67959 | 461 | | 6115 | 16953 | 325 | | 22743 | 54557 | 1556 | | 5400 | 18195 | 604 | |
| Currently married | 2386 | 3157 | 27 | | 12117 | 42123 | 313 | | 7912 | 14870 | 113 | | 19855 | 7196 | 444 | |
| Formerly married | 586 | 885 | 6 | | 3689 | 10423 | 88 | | 2020 | 3197 | 35 | | 7614 | 23063 | 244 | |
| Ever married* | 23 | 22 | 0 | | 447 | 1194 | 13 | | 488 | 792 | 12 | | 2524 | 10195 | 121 | |
| Not known | 5418 | 3588 | 46 | | 511 | 1077 | 24 | | 5402 | 3807 | 127 | | 915 | 1888 | 56 | |

**Table 2 (continued) | Description of study population and prevalence of potential risk factors**

| | Men | | | | | | | | Women | | | | | | | |
|---|---|---|---|---|---|---|---|---|---|---|---|---|---|---|---|---|
| | 15-24 | | | | 25-49 | | | | 15-24 | | | | 25-49 | | | |
| | N | PY | S | M | N | PY | S | M | N | PY | S | M | N | PY | S | M |
| **Number of partners in the last year** | | | | | | | | | | | | | | | | |
| No partners | 12661 | 15327 | 102 | | 1868 | 1994 | 17 | | 11231 | 13205 | 302 | | 4123 | 5133 | 58 | |
| 1 partner | 6870 | 7540 | 78 | | 8747 | 12153 | 133 | | 11630 | 14232 | 541 | | 17612 | 29369 | 376 | |
| 2 partners | 1948 | 1778 | 25 | | 2995 | 3604 | 45 | | 467 | 364 | 21 | | 558 | 527 | 16 | |
| 3+ partners | 996 | 865 | 10 | | 1392 | 1523 | 26 | | 73 | 58 | 2 | | 106 | 93 | 2 | |
| Don't know | 26045 | 50102 | 327 | | 17991 | 52496 | 541 | | 28818 | 49363 | 979 | | 29208 | 89413 | 1019 | |
| **Had more than one partner in the last year** | | | | | | | | | | | | | | | | |
| No | 17191 | 22996 | 181 | | 10415 | 14374 | 151 | | 20539 | 27607 | 843 | | 20979 | 34943 | 438 | |
| Yes | 2754 | 2643 | 34 | | 3900 | 5127 | 71 | | 535 | 423 | 22 | | 649 | 620 | 18 | |
| Don't know | 26032 | 49973 | 325 | | 17988 | 52269 | 540 | | 28803 | 49194 | 978 | | 29200 | 88973 | 1015 | |
| **Had a casual partner in the last year** | | | | | | | | | | | | | | | | |
| No | 13985 | 17998 | 167 | | 7867 | 10092 | 121 | | 15900 | 20481 | 733 | | 14862 | 21664 | 323 | |
| Yes | 2602 | 2563 | 35 | | 2143 | 2623 | 44 | | 2196 | 2143 | 98 | | 2055 | 2634 | 43 | |
| Don't know | 26535 | 55051 | 339 | | 18347 | 59055 | 597 | | 29296 | 54599 | 1012 | | 29642 | 100237 | 1104 | |
| **Had a regular partner in the last year** | | | | | | | | | | | | | | | | |
| No | 12532 | 15225 | 132 | | 7074 | 9151 | 74 | | 13835 | 15833 | 421 | | 13433 | 18953 | 174 | |
| Yes | 4660 | 5336 | 70 | | 2932 | 3565 | 91 | | 5430 | 6791 | 410 | | 3709 | 5346 | 193 | |
| Don't know | 26535 | 55051 | 339 | | 18347 | 59055 | 597 | | 29296 | 54599 | 1012 | | 29642 | 100237 | 1104 | |
| **Age difference(s) with partners in the last year** | | | | | | | | | | | | | | | | |
| within 5 years of age | 5744 | 6307 | 69 | | 4738 | 5226 | 74 | | 6013 | 6491 | 274 | | 7467 | 9512 | 152 | |
| 5+ years difference | 911 | 758 | 8 | | 3589 | 3877 | 46 | | 3195 | 3094 | 125 | | 5565 | 6132 | 78 | |
| 10+ years difference | 65 | 51 | 0 | | 2109 | 2375 | 22 | | 989 | 872 | 36 | | 3309 | 3643 | 39 | |
| Age unknown | 1335 | 1284 | 24 | | 1063 | 987 | 15 | | 1560 | 1490 | 86 | | 2417 | 2631 | 63 | |
| No partner | 12574 | 15192 | 101 | | 1514 | 1671 | 15 | | 11065 | 13037 | 295 | | 3661 | 4657 | 51 | |
| No data | 26157 | 52020 | 339 | | 18233 | 57635 | 590 | | 28995 | 52241 | 1028 | | 29539 | 97960 | 1088 | |
| **Prevalence of untreated infection in potential opposite sex partners** | | | | | | | | | | | | | | | | |
| | | 146487 | | 4.8 | | 142049 | | 11.7 | | 152593 | | 5.5 | | 246731 | | 11.0 |
| **Had a new partner in the last year** | | | | | | | | | | | | | | | | |
| No new partners | 14220 | 17919 | 163 | | 8910 | 11506 | 143 | | 16799 | 20946 | 672 | | 16569 | 23601 | 320 | |
| New partner(s) | 5170 | 5253 | 61 | | 2950 | 2878 | 53 | | 3110 | 2708 | 149 | | 1641 | 1430 | 51 | |
| No data | 26317 | 52440 | 317 | | 18207 | 57387 | 566 | | 29183 | 53569 | 1023 | | 29629 | 99505 | 1100 | |
| **Coital frequency with all partners in the last year** | | | | | | | | | | | | | | | | |
| Zero | 132 | 99 | 0 | | 254 | 183 | 2 | | 191 | 142 | 2 | | 726 | 681 | 6 | |
| <1/week | 456 | 364 | 0 | | 640 | 482 | 3 | | 473 | 345 | 3 | | 1273 | 959 | 11 | |
| 1+ times/week | 599 | 481 | 2 | | 2220 | 2151 | 22 | | 1174 | 838 | 8 | | 3922 | 3748 | 33 | |
| No data | 28900 | 74668 | 538 | | 18962 | 68954 | 735 | | 31868 | 75897 | 1832 | | 30590 | 119148 | 1422 | |
| **Is circumcised (men only)** | | | | | | | | | | | | | | | | |
| Not | 5387 | 12921 | 72 | | 2341 | 7777 | 58 | | | | | | | | | |
| Circumcised | 4819 | 10704 | 18 | | 3760 | 13074 | 85 | | | | | | | | | |
| no information | 21236 | 51987 | 451 | | 14182 | 50920 | 619 | | 31924 | 77224 | 1844 | | 30701 | 124536 | 1471 | |

**Table 2 (continued) | Description of study population and prevalence of potential risk factors**

| | Men | | | | | | | | Women | | | | | | | |
|---|---|---|---|---|---|---|---|---|---|---|---|---|---|---|---|---|
| | 15-24 | | | | 25-49 | | | | 15-24 | | | | 25-49 | | | |
| | N | PY | S | M | N | PY | S | M | N | PY | S | M | N | PY | S | M |
| **Condoms used consistently with all partners in the last year** | | | | | | | | | | | | | | | | |
| No partners | 15608 | 17627 | 111 | | 4097 | 3754 | 38 | | 14543 | 15734 | 328 | | 8549 | 8715 | 100 | |
| No condom use | 4967 | 4885 | 64 | | 8480 | 10979 | 120 | | 7215 | 7207 | 237 | | 14960 | 20724 | 206 | |
| Some condom use | 3874 | 4314 | 61 | | 3054 | 3567 | 74 | | 4524 | 5099 | 311 | | 3520 | 4012 | 152 | |
| Consistent use | 852 | 893 | 5 | | 352 | 364 | 4 | | 583 | 569 | 9 | | 398 | 530 | 4 | |
| Insufficuent data | 935 | 774 | 5 | | 1694 | 1380 | 9 | | 957 | 733 | 3 | | 2334 | 1816 | 13 | |
| Not asked | 25864 | 47119 | 295 | | 18004 | 51726 | 516 | | 28658 | 47882 | 955 | | 29268 | 88740 | 997 | |
| **Ended a partnership in the last year** | | | | | | | | | | | | | | | | |
| No | 15562 | 21373 | 209 | | 10010 | 14732 | 193 | | 17950 | 23868 | 818 | | 18258 | 28739 | 400 | |
| Yes | 2416 | 2247 | 21 | | 1983 | 1956 | 31 | | 1061 | 877 | 41 | | 926 | 839 | 21 | |
| Don't know | 26275 | 51991 | 311 | | 18092 | 55083 | 538 | | 29090 | 52478 | 984 | | 29385 | 94958 | 1050 | |

Mean across imputations of: numbers of people (N), person years (PY) and seroconversions (S), by sex and age, in each study and in the pooled dataset, for each categorical risk factor and means (M) for continuous variables. The table includes only data which contributed to the survival analysis.

Numbers of people (N), person-years of follow up (PY) and seroconversions (S) observed in the incidence cohort by sex by sex and age between 2005-16 among 15-49 year olds from all studies. NB. For some variables, the number of individuals sums to more than the total number in the incidence cohort because an individual may feature in more than one category during the period of follow up. Some small fluctuations in the total numbers across the different risk factors (e.g., plus or minus one seroconversion) arises due to rounding differences when taking the mean across imputations. *In one survey, respondents were asked only whether they had ever been married and not for their current marital status.

her number of partners that was seen in the crude did not persist in the adjusted analysis. In the crude analysis there was a suggestion that HIV acquisition hazard increased as the age gaps between partners increased but this may have been due to chance and there was no evidence for this association in the adjusted model.

### Older women

For older women, having moved house in the last year was associated with increased risk of HIV acquisition (adj HR 1.50 95% CI 1.19-1.88). Unlike for younger people, among older women secondary education was associated with a higher chance of HIV acquisition (adj HR 1.25 95% CI 1.01-1.53). The risk of HIV acquisition in older women who were married was around half that experienced by those never or formerly married. The association with new partnerships remained for older women (adj HR 1.87 95% CI 1.29-2.71). Inconsistent condom use was associated with acquisition of infection among older women (adj HR 2.06, 95% CI 1.59-2.67).

In the crude analysis, there was a trend of increasing HIV acquisition hazard with higher number of partners which was not seen in the adjusted model. There was weak evidence that older women with a larger age gap between their partners had lower hazards of HIV acquisition (crude HR 0.65 95% CI 0.41-1.01), the opposite effect to that observed among young women, but this did not persist after adjustment for other characteristics of the partners.

### Associations between community level factors and HIV incidence

Community rates of partner acquisition (CPAR) were available in Karonga, Kisesa and Rakai, and were highest for young men, and higher for men than for women. Community partner loss rates (CPLR) were lower than acquisition rates, except for older men (Table 2) and were not associated with incidence. CPAR among peers showed a crude association with incidence for men and older women (Table 3) and men's PAR was associated with risk for older women. In the adjusted models, men's PAR remained associated with HIV acquisition for older men and older women (Supplementary Data 11 and 12). The effect of a one percentage point in the men's partner acquisition rate was a 3% increase in the HIV incidence hazard for both older women and older men (adjusted HRs 1.03, 95% CI 1.004-1.05 and 1.03, 95% CI 1.01-1.04, respectively).

### Sensitivity analyses

To assess whether the results were unduly influenced by any one study, we repeated the models above for each study and did a meta-analysis to combine the estimates leaving one study out each time. The results in Supplementary Table 2 show the estimates obtained are not determined by any single study.

To see if the risk factors were different in the most recent period we repeated the preferred models for data from 2013-16 only and restricted to the studies which had data for this period: Kisesa, Masaka, Rakai, uMkhanyakude and Kisumu (Supplementary Data 13). For older men, the HR were similar to the model using all years but the confidence intervals were wider. New partners did not show an association for older women but other results were similar. For young men and women the only difference was that the confidence intervals for casual and regular partner estimates widened to include one.

## Discussion

This multi-country study combined eight open cohort studies that followed almost 100,000 people for 349,140 person years, observed 4,617 new infections and collected detailed demographic and behavioural data. This large harmonised dataset facilitated an extensive exploration of risk factors for HIV incidence in the general population.

For young women, former marriage and having regular, casual and new partners were associated with increased risk of acquiring HIV

**Table 3 | Crude HR and 95% CI for HIV seroconversion in pooled data, by age and sex, adjusted for study**

| Risk Factor | Young Men | Older Men | Young Women | Older Women |
|---|---|---|---|---|
| Incidence rate/1000 person years | 6.7 (6.2-7.3) | 10.7 (10.0-11.4) | 23.3 (22.3-24.3) | 11.9 (11.3-12.4) |
| **Education** | | | | |
| No formal education | 1.66 (0.45-6.09) | 1.39 (0.76-2.56) | 1.32 (0.53-3.29) | 1.09 (0.75-1.60) |
| Primary | 1 | 1 | 1 | 1 |
| Secondary | 0.71 (0.48-1.03) | 1.20 (0.96-1.51) | **0.74 (0.59-0.92)** | **1.29 (1.05-1.59)** |
| Tertiary | 1.21 (0.40-3.61) | 0.96 (0.61-1.48) | 0.36 (0.10-1.34) | 0.62 (0.37-1.02) |
| Unknown | **0.75 (0.58-0.96)** | **1.45 (1.20-1.74)** | 0.98 (0.83-1.15) | 0.91 (0.73-1.12) |
| **Urban or rural Resident** | | | | |
| Rural | 1 | 1 | 1 | 1 |
| Urban | **1.79 (1.13-2.83)** | 1.29 (0.82-2.01) | 1.18 (0.88-1.57) | 1.35 (0.99-1.85) |
| Peri-urban | **1.69 (1.36-2.10)** | **2.80 (2.24-3.51)** | **1.25 (1.11-1.40)** | **1.29 (1.10-1.51)** |
| <1 km from road | 1.03 (0.29-3.58) | 0.58 (0.33-1.01) | 1.28 (0.73-2.27) | 1.29 (0.83-2.02) |
| >1 km from road | 1.02 (0.32-3.21) | **0.35 (0.19-0.65)** | **0.39 (0.16-0.94)** | **0.39 (0.21-0.75)** |
| agricultural estate | 1.52 (0.22-10.52) | **1.90 (1.29-2.80)** | **2.53 (1.17-5.49)** | **1.91 (1.19-3.04)** |
| commercial center | 3.08 (0.81-11.71) | **1.95 (1.31-2.92)** | **3.47 (1.85-6.52)** | **3.16 (2.11-4.71)** |
| roadside trading | **2.79 (1.01-7.73)** | 1.04 (0.60-1.79) | **2.61 (1.46-4.67)** | **1.76 (1.18-2.62)** |
| subsistence farming | 1.94 (0.49-7.72) | 0.98 (0.56-1.73) | 1.55 (0.73-3.29) | 1.23 (0.77-1.96) |
| **Moved house in the last year** | | | | |
| No | 1 | 1 | 1 | 1 |
| Yes | **1.68 (1.11-2.55)** | 1.30 (0.94-1.81) | **1.29 (1.04-1.60)** | **1.84 (1.47-2.31)** |
| **Calendar year (grouped)** | | | | |
| 2005-08 | 1 | 1 | 1 | 1 |
| 2009-12 | 0.99 (0.77-1.26) | **0.81 (0.66-0.99)** | 0.90 (0.78-1.03) | 0.86 (0.70-1.05) |
| 2013-16 | **0.67 (0.50-0.89)** | **0.64 (0.51-0.80)** | **0.70 (0.61-0.81)** | **0.60 (0.46-0.79)** |
| 2005-08 uMkhanyakude | | | | **0.80 (0.64-0.99)** |
| 2009-12 uMkhanyakude | | | | 0.93 (0.75-1.16) |
| **Partner acquisition rate among peers[1]** | | | | |
| | **1.01 (1.00-1.03)** | **1.03 (1.01-1.04)** | 0.97 (0.94-1.01) | **1.04 (1.00-1.09)** |
| **Partner loss rate among peers[1]** | | | | |
| | 1.01 (0.99-1.04) | 1.00 (0.99-1.00) | 1.04 (0.96-1.12) | 1.03 (0.96-1.10) |
| **Partner acquisition rate in potential opposite sex partners[1]** | | | | |
| | 0.90 (0.74-1.09) | **1.08 (1.03-1.14)** | 1.00 (0.98-1.02) | **1.03 (1.01-1.04)** |
| **Partner loss rate in potential opposite sex partners[1]** | | | | |
| | 0.82 (0.59-1.14) | **1.15 (1.01-1.30)** | 1.01 (0.99-1.03) | 1.00 (0.99-1.01) |
| **Current marital status** | | | | |
| Never married | **0.32 (0.20-0.51)** | **2.67 (2.26-3.15)** | 1.05 (0.81-1.36) | **2.51 (1.96-3.22)** |
| Currently married | 1 | 1 | 1 | 1 |
| Formerly married | 1.60 (0.55-4.68) | **1.47 (1.12-1.94)** | **4.15 (2.61-6.60)** | **2.13 (1.79-2.53)** |
| Ever married no current info | ** | 1.45 (0.78-2.68) | 1.11 (0.56-2.18) | 0.86 (0.64-1.15) |
| Not known | **0.45 (0.25-0.81)** | **3.05 (1.87-4.97)** | 0.96 (0.69-1.34) | **2.21 (1.51-3.22)** |
| **Number of partners in the last year** | | | | |
| No partners | **0.62 (0.43-0.91)** | 0.75 (0.39-1.46) | **0.51 (0.43-0.60)** | 0.85 (0.60-1.22) |
| 1 partner | 1 | 1 | 1 | 1 |
| 2 partners | 1.71 (0.94-3.10) | 1.15 (0.75-1.75) | **2.36 (1.40-3.97)** | **3.15 (1.69-5.88)** |
| 3+ partners | 1.82 (0.77-4.27) | 1.55 (0.94-2.56) | ** | ** |
| No data | 0.96 (0.68-1.34) | 0.94 (0.74-1.21) | **0.72 (0.63-0.83)** | 0.93 (0.80-1.08) |
| **Had more than one partner in the last year** | | | | |
| No | 1 | 1 | 1 | 1 |
| Yes | **2.32 (1.46-3.70)** | 1.32 (0.94-1.86) | **3.03 (1.83-5.02)** | **3.15 (1.77-5.60)** |
| No data | 1.27 (1.00-1.61) | 0.98 (0.78-1.23) | 0.97 (0.85-1.10) | 0.94 (0.81-1.09) |
| **Had a casual partner in the last year** | | | | |
| No | 1 | 1 | 1 | 1 |
| Yes | 1.42 (0.91-2.22) | 1.38 (0.91-2.11) | **1.57 (1.22-2.02)** | 1.27 (0.86-1.88) |
| No data | 1.25 (0.99-1.59) | 0.84 (0.66-1.07) | 0.99 (0.87-1.13) | 0.93 (0.79-1.09) |

**Table 3 (continued) | Crude HR and 95% CI for HIV seroconversion in pooled data, by age and sex, adjusted for study**

| Risk Factor | Young Men | Older Men | Young Women | Older Women |
|---|---|---|---|---|
| **Had a regular partner in the last year** | | | | |
| No | 1 | 1 | 1 | 1 |
| Yes | **1.55 (1.09-2.22)** | **3.17 (2.18-4.59)** | **1.75 (1.48-2.08)** | **2.34 (1.82-3.01)** |
| No data | **1.35 (1.05-1.74)** | 1.25 (0.93-1.69) | **1.20 (1.02-1.41)** | **1.30 (1.06-1.59)** |
| **Age difference(s) with partners in the last year** | | | | |
| Partner within 5 years of age | 1 | 1 | 1 | 1 |
| Partner 5+ years older/younger | 1.29 (0.44-3.79) | 0.85 (0.55-1.31) | 1.08 (0.84-1.40) | 0.81 (0.58-1.11) |
| Partner 10+ years older/younger | ** | 0.65 (0.36-1.16) | 1.39 (0.90-2.14) | 0.65 (0.41-1.01) |
| Age not known/not given | 1.01 (0.58-1.79) | 1.07 (0.54-2.11) | 1.11 (0.83-1.47) | 1.07 (0.75-1.52) |
| No partner in ref period | **0.57 (0.39-0.83)** | 0.63 (0.31-1.27) | **0.52 (0.43-0.63)** | 0.69 (0.46-1.03) |
| No data | 0.89 (0.64-1.24) | **0.73 (0.54-0.98)** | **0.76 (0.64-0.89)** | **0.78 (0.63-0.96)** |
| **Prevalence of untreated infection in potential opposite sex partners** | | | | |
| | **1.07 (1.05-1.09)** | **1.06 (1.05-1.07)** | **1.02 (1.01-1.03)** | **1.05 (1.03-1.06)** |
| **Had a new partner in the last year** | | | | |
| No | 1 | 1 | 1 | 1 |
| Yes | **1.49 (1.03-2.15)** | **1.48 (1.00-2.19)** | **1.80 (1.44-2.24)** | **2.52 (1.75-3.64)** |
| No data | **1.29 (1.00-1.65)** | 0.79 (0.63-1.00) | 1.06 (0.92-1.21) | 0.94 (0.80-1.10) |
| **Coital frequency with all partners in the last year** | | | | |
| Zero | ** | ** | ** | 0.58 (0.13-2.64) |
| <1/week | 1 | 1 | 1 | 1 |
| 1+ times/week | ** | 1.65 (0.28-9.63) | 1.10 (0.15-7.86) | 0.74 (0.29-1.92) |
| No data | ** | 1.77 (0.34-9.29) | 1.27 (0.22-7.32) | 0.81 (0.35-1.85) |
| **Is circumcised** | | | | |
| Not | 1 | 1 | | |
| Circumcised | **0.36 (0.20-0.66)** | 0.88 (0.61-1.27) | | |
| no information | **1.36 (1.04-1.77)** | **1.65 (1.23-2.22)** | | |
| **Condoms used consistently with all partners in the last year** | | | | |
| No partners | **0.57 (0.40-0.83)** | 0.93 (0.59-1.47) | **0.56 (0.46-0.68)** | 1.25 (0.93-1.69) |
| No condom use | 1 | 1 | 1 | 1 |
| Some condom use (sometimes or some partners | 0.86 (0.54-1.36) | **1.91 (1.35-2.69)** | 1.11 (0.90-1.37) | **2.56 (1.99-3.28)** |
| Complete condom use (all partners=always) | 0.83 (0.23-2.92) | 1.05 (0.31-3.55) | 1.19 (0.49-2.92) | 0.98 (0.24-4.08) |
| Not sure- 6+ptnrs or missing data | 1.95 (0.57-6.65) | 0.56 (0.22-1.43) | ** | 1.13 (0.54-2.38) |
| No data | 0.82 (0.58-1.14) | 0.92 (0.71-1.18) | **0.74 (0.62-0.89)** | 1.18 (0.99-1.41) |
| **Ended a partnership in the last year** | | | | |
| No | 1 | 1 | 1 | 1 |
| Yes | 1.30 (0.76-2.21) | 1.16 (0.75-1.78) | **1.64 (1.15-2.35)** | 1.44 (0.91-2.26) |
| No data | 1.20 (0.95-1.52) | **0.75 (0.61-0.92)** | 0.97 (0.85-1.10) | 0.94 (0.81-1.10) |

Bold font indicates estimates where the confidence intervals do not include 1.
** CI not estimated due to zero failures in some imputations.
[1]Estimated for Karonga, Kisesa and Rakai only.

while secondary education was protective. For older women, being unmarried, having changed residence in the last year, having a new partner and inconsistent condom use were associated with risk and tertiary education was protective. For young men, never being married and being circumcised were associated with lower risk and no behavioural risk factors were identified. In contrast, there were few protective factors for older men, among whom former marriage and having a regular partner increased the risk. The community prevalence of untreated HIV infection was associated with HIV acquisition among men and women in both age groups. The rate of partner acquisition by men in the community increased the risk of HIV acquisition among older men and women. The relative paucity of identified risk factors for men compared to women suggests that there were either unmeasured factors which affect HIV incidence among men or the correlation between reported behaviour and risk was weaker among men. For young men, the strong association with the prevalence of untreated infection, combined with the lack of behavioural risk factors may

indicate that their risk is affected by characteristics of partners besides their number and relationship. A similar pattern of association by sex was observed in the PopART trial in Zambia and South Africa[36]

We found associations that were consistent across the different studies and also observed variations, by sex, age and study in the prevalence of risk factors. We have added information for populations in Malawi where risk factors in the general population have not previously been described. All of these had been identified as risks by other studies, and many have been used as components of HIV risk scoring algorithms but this is the first time they have been compared in a single analysis and shown to be consistently associated with risk in different populations[37]. The effects of some risk factors varied by age and/or sex, which matters where prevention activities are targeted a priori by age and sex. Former marriage was a risk for all except young men. Only the prevalence of untreated infection was associated with acquiring infections for both men and women and for both age groups, as seen elsewhere[38,39]. New partnerships were a risk for women but not men.

**Table 4 | Results from models of HIV incidence for each age and sex group, for 2005-16 inclusive, showing adjusted HR and 95% CI**

| | Young Men | | Older Men | | Young Women | | Older Women | |
|---|---|---|---|---|---|---|---|---|
| | Adj HR | 95% CI | Adj HR | 95% CI | Adj HR | 95% CI | Adj HR | 95% CI |
| **Education** | | | | | | | | |
| No formal education | 1.19 | 0.32-4.43 | | | 1.26 | 0.50-3.15 | 1.09 | 0.74-1.60 |
| Primary | 1 | | | | 1 | | 1 | |
| Secondary | 0.72 | 0.49-1.06 | | | **0.70** | **0.56-0.87** | **1.25** | **1.01-1.53** |
| Tertiary | 1.08 | 0.36-3.24 | | | 0.33 | 0.09-1.20 | 0.60 | 0.36-1.00 |
| Unknown | 0.64 | 0.49-0.83 | | | 0.87 | 0.74-1.04 | 0.87 | 0.70-1.09 |
| **Marital Status** | | | | | | | | |
| Never married | **0.38** | **0.23-0.61** | 1.12 | 0.85-1.47 | 1.14 | 0.87-1.49 | **2.30** | **1.78-2.97** |
| Currently married | 1 | | 1 | | 1 | | 1 | |
| Formerly married | 1.41 | 0.50-3.99 | **1.41** | **1.06-1.86** | 2.71 | 1.63-4.50 | 1.95 | 1.62-2.34 |
| Ever married (no current info) | | | **0.48** | **0.25-0.94** | 1.07 | 0.54-2.13 | 0.94 | 0.70-1.27 |
| Not known | 0.62 | 0.34-1.15 | 1.02 | 0.59-1.78 | 1.08 | 0.76-1.53 | 2.12 | 1.44-3.12 |
| **Changed residence within the last year** | | | | | | | | |
| No | | | | | | | 1 | |
| Yes | | | | | | | **1.50** | **1.19-1.88** |
| **Is circumcised (men only)** | | | | | | | | |
| No | 1 | | | | | | | |
| Yes | **0.38** | **0.21-0.70** | | | | | | |
| Not known | 1.25 | 0.95-1.63 | | | | | | |
| **Regular partner(s) in last year** | | | | | | | | |
| None | | | 1 | | 1 | | | |
| One or more | | | **1.82** | **1.24-2.68** | **1.70** | **1.39-2.08** | | |
| Not known | | | 1.18 | 0.86-1.62 | 0.98 | 0.69-1.40 | | |
| **New partnerships in last year** | | | | | | | | |
| None | | | | | 1 | | 1 | |
| One or more | | | | | **1.35** | **1.07-1.72** | **1.87** | **1.29-2.71** |
| Not known | | | | | 1.36 | 0.96-1.93 | 1.11 | 0.81-1.51 |
| **Casual partners in last year** | | | | | | | | |
| None | | | | | 1 | | | |
| One or more | | | | | **1.82** | **1.38-2.39** | | |
| Not known | | | | | 1.000 | | | |
| **Condom use in last year** | | | | | | | | |
| No partners | | | | | | | 1.09 | 0.77-1.54 |
| No condom use | | | | | | | 1 | |
| Some condom use | | | | | | | **2.06** | **1.59-2.67** |
| Consistent condom use | | | | | | | 0.69 | 0.17-2.90 |
| Not known | | | | | | | 1.10 | 0.81-1.49 |
| **Prevalence untreated HIV in potential opposite sex partners** | **1.07** | **1.04-1.09** | **1.02** | **1.00-1.03** | **1.01** | **1.00-1.02** | **1.03** | **1.02-1.04** |
| **Study** | | | | | | | | |
| Karonga | 0.35 | 0.09-1.29 | 0.56 | 0.33-0.95 | 0.56 | 0.31-1.02 | 0.54 | 0.34-0.85 |
| Kisesa | 1.48 | 0.62-3.57 | 1.12 | 0.75-1.67 | 1.25 | 0.76-2.07 | 1.07 | 0.73-1.57 |
| Manicaland | 1.31 | 0.52-3.30 | 1.84 | 1.22-2.76 | 2.62 | 1.58-4.32 | 1.01 | 0.67-1.54 |
| Masaka | 1 | | 1 | | 1 | | 1 | |
| Rakai | 2.37 | 1.13-4.98 | 1.24 | 0.90-1.72 | 2.13 | 1.42-3.20 | 1.21 | 0.86-1.69 |
| uMkhanyakude | 6.95 | 3.52-13.71 | 2.98 | 1.79-4.97 | 9.64 | 6.84-13.57 | 2.37 | 1.66-3.39 |
| Kisumu | 1.51 | 0.72-3.17 | 0.87 | 0.59-1.29 | 1.07 | 0.71-1.60 | 1.13 | 0.82-1.55 |
| Ifakara | 0.56 | ** | 0.87 | 0.27-2.78 | 2.37 | 1.02-5.53 | 1.52 | 0.83-2.80 |
| *Number of seroconversions†* | 537 | | 760 | | 1842 | | 1469 | |
| *Number of respondent†s* | 28348 | | 19037 | | 31895 | | 30689 | |
| *Total person years†* | 73763 | | 71528 | | 76838 | | 124240 | |

Bold font highlights estimates whose confidence interval does not include 1.

** CI not estimated due to zero failures in some imputations.

† Mean across imputation.

The effect of CPAR on incidence shows that other people's behaviour can have a demonstrable effect on individual risk. This has also been reported elsewhere with a correlation between CPAR and HIV prevalence seen using national level data and results from the PopART trial showed men's behaviour at the community level to be predictive of incidence among women[36,40]. Communities with higher CPAR will have a denser sexual network, with new partnerships more closely spaced in time, increasing STI risk[41,42].

We did not find associations with multiple partners, which had been seen in other studies[6,9,14,18,20–22,26,28,43]. This may be because the risk of multiple partnerships is experienced at the beginning of the most recently acquired partnership and we controlled for this with variables describing new and casual partnerships[11,14,18,20–22,26,28,29].

Whilst effective against HIV infection, condom use was difficult to incorporate into these risk factor models, partly because it was not common. In an analysis of the general population, where most people have very low levels of HIV risk, condom use can be a marker of risky behaviour even in a generalised epidemic setting. Our results indicate condom use reported by older women was insufficient to offset the increased risk they experienced. It suggests that at least one person in the partnership was aware of a risk of HIV transmission and the need for prophylaxis, which could make members of this group open to using PrEP. However, the older women may not be the instigators of condom use, or in a position to obtain and use PrEP and they would also be protected by effective treatment of the male partners.

We did not find universal associations between different types of partnership and acquisition of infection. This may be an artefact of our analytical approach. We had a separate category for new partnership, which encompassed many of the casual partnerships and was generally predictive of incidence. Fewer people were in casual partnerships than regular so power to detect an association was lower. Semantic differences in the terms used for partner type, due to translation and mapping local descriptions onto standard terms, might have led to some misclassification of partner types.

This highlights a problem of labelling particular types of partnership as those which present a greater risk of HIV transmission. Partnerships frequently evolve over time, and what begins as casual (and new) can progress to something regular and onwards to marriage. This complicates assessment of the effect of condom use. Where condoms are used, it is most often at the start of the partnership and they are frequently discontinued as the relationship matures, particularly if childbearing is intended. If couples do not test and disclose before discontinuing condom use then adequate condom use in the initial, casual, stage may postpone HIV transmission until later, when the partnership has become more stable. We know that condom use levels vary by type of partnership and we observed differences by study, age and sex (Table 1 and Supplementary Data 1 and 2) so these might contribute to the differences in partnership risks[44]. The level of risk in different partnerships may be further modified by patterns of partner acquisition for people in partnerships wherein concordant negative partnerships can become serodiscordant. The number and type of existing partners affect the chances of acquiring a new partner. People with partners are less likely than single people to acquire a new partner and the extent of the difference varies by sex, country and the type of partnership[40]. Differences in the extent of mutual monogamy between marital, regular and casual partnerships will affect incidence risk, unless this is offset by condom use or PrEP. Our analyses were adjusted for partner acquisition and condom use at the individual-level but we did not capture population-level differences in partnership characteristics. Selection effects caused by differences in partnership behaviours might also explain the different effects of circumcision for younger and older men. Our results suggest that older men who are circumcised have a different risk profile to younger men, one that has not been captured in our analysis. Additionally, we were unable to distinguish medical from traditional circumcision, and older men may

be more likely that younger men to have undergone a procedure that is less effective for HIV prevention.

The importance of new partners and the largely protective effect of marriage, also seen by others[6,9,10,15,18,21,22,28,43], implies that people navigate periods of increased risk before arriving into the relative safety of a marriage. The protective effect of marriage may be overstated by this, and other, similar analyses. Discordant couples who have sex before marriage and do not take steps to prevent transmission of infection are likely to become seroconcordant. Marriages that start out, or become serodiscordant, will likewise remain serodiscordant for only a short period in the absence of prevention measures. Work from Rakai showed that a large proportion of new infections could be linked to an incident infection in a household contact[7]. Therefore, people included in an incidence analysis (who are HIV-negative) and initially observed while currently married are likely to have an HIV-negative partner or be taking effective steps to prevent HIV transmission from a positive partner. We considered looking only at recent marriages to investigate this but did not have sufficient statistical power.

The prevention response to any association with partner type would be similar: the use of condoms or PrEP, testing and disclosure and effective treatment for PLHIV. Our results show that using one universal definition of a 'risky' type of partnership as the trigger for intervention could fail to capture people before they are exposed to risk. It could be more useful to think about how people navigate transitions in their partnerships and what is needed to make these transits safe, rather than focussing on people who have already spent some time in a particular category of partnership. Finding out how to identify people who are about to make risky transitions could be useful, for example, the substantial increase in risk associated with a new partnership would be best addressed before the person entered into the partnership, not after they have done so.

Harmonisation and secondary analysis of disparate sources of data inevitably introduces some limitations and can be especially complicated for sexual behaviour, where reporting is subjective and may be affected by social desirability bias. Different survey frequencies and incomplete participation in every survey round led to person time being classified as "unknown" for many aspects of behaviour and the extent of this varied between studies and over time. The different sizes and durations of the studies meant some contributed more data than others. These differences could have affected the results as availability of data was correlated with the study and calendar time, which were also associated with HIV incidence. We mitigated against this by including a fixed effect for study in all models and by conducting a sensitivity analysis. In some models, some of the "unknown" categories retained an association with HIV incidence, possibly because people with missing data were different (e.g., due to work or travel patterns) or because there was residual confounding by study or calendar period. Sensitivity analysis showed the results were robust to the omission of each study, and there was not sufficient study heterogeneity to vitiate the pooled estimates. Our population-level measures were estimated based on data for the entire study area, which may have masked geographic heterogeneity in risk. Without good information on the extent of mixing between residents of different areas of the studies it was not possible to produce estimates that were meaningfully geographically stratified. By using the average across the whole area our measures may be less precise but, given the small size of most of the studies, and the detection of plausible associations we believe this to be a minor limitation.

Using harmonised data meant the omission of some information on wealth, alcohol use and sexual practices which have been shown to have important associations with HIV acquisition[3,6,11,15,20,21,26,33,36,43]. We do not think their omission has affected our results. Wealth is difficult to harmonise as the locally relevant measures are not easy to translate across contexts. We did not attempt to do this because we expected

the effects of wealth on behaviour, and thus HIV risk, to differ between studies, as has been seen elsewhere[40]. Most studies had some information on alcohol use and specific sexual practices but these were for different time periods. They are very proximate factors which may change with each encounter and some of the information is imprecise, which introduces noise in the associations when they are included in regression analysis of whole populations over a long timescale. Proximate measures such as these, and condom use and PrEP, are difficult to capture at this scale. There was little PrEP available in the the study populations at the time of data collection. How the availability of PrEP has affected the associations we have shown will depend on how these are correlated with access to and uptake of PrEP. Particularly for women our risk factors speak of periods when women might have less agency and power; in those moments PrEP could be transformative and might remove that excess risk, but those same factors might also hinder its use.

The socially-conditioned and subjective nature of self-reported data on sexual behaviour is a perennial problem. We believe these data to be high-quality because there are high levels of trust between the study communities and researchers in these long-running projects and, because each study is bespoke to a small area, the enquiries are tailored to the local population[45]. Differences in the local meaning ascribed to descriptions of partners, alongside social desirability bias, may have led to the quite large differences in the prevalence of reported partner types. Finding coherent associations between several hypothesised risk factors and HIV incidence shows the measures, even if inaccurate, are capturing differences in HIV risk.

The associations identified here related to new infections that occurred between 2005 and 2016 and the risks we have identified may be less important today. Treatment was available for this entire period though availability was limited in some studies before 2010 and treatment eligibility expanded over time. Sensitivity analysis limited to 2013-16 data suggests that the risk factors in that period were similar to those identified for the whole period. Many were identified as risks in the late 1990s and early 2000s, at very different phases of the epidemic. This consistency over time could mean that they remain important going forwards.. An analysis of slightly more recent PHIA data found only two risk factors for women, living in a subnational area with high HIV-1 viremia and having a non-cohabiting partner but this was based on a small number of recent infections which may have precluded identification of other risks[46]. There are no studies with recently collected data on HIV risk factors and their association with documented HIV seroconversion in the general population in sub-Saharan Africa. This is likely explained by changing research priorities and funding, meaning more recent data are not available for most of the ALPHA cohorts, and due to the length of time it takes to harmonise data for multicountry studies. Initiatives such as the African Population Cohorts Consortium will increase the speed with which results can be produced[47].

We observed a large number of respondents, but they came from just eight districts in six countries so may not represent the experience elsewhere in these countries, or in other countries in the region. That the results are stable across the different studies, which represent a diverse range of peoples, gives us confidence that these findings can be more widely applicable.

Our results raise the possibility that the timing of prevention intervention relative to behaviour might be important. The importance of new partnerships and former marriage suggests that the liminal periods around partnerships' beginning and end could be the start of periods of increased risk. The identification of community level risk factors risk point to the utility of using community characteristics to target prevention, perhaps before individual behaviour escalates risk, and even of altering these characteristics to ameliorate that increased risk. Other authors have shown substantial age and spatial variation in the prevalence of risk factors[48], that people cycle in and out of periods

of increased risk[49], and the utility of community level measures to target prevention[46] and these findings reinforce theirs. This could be worthwhile in contexts, such as mature generalised epidemic settings, where the "high risk" behaviours are not atypical behaviour but things most people experience or do at some point. Our measures of both untreated prevalence and partner acquisition can be calculated from surveys, such as DHS. Continued effective prevention, in the face of falling incidence, requires better targeting to reach those most in need whilst remaining cost effective and these could be two new approaches to improve that targeting. Demographic and social changes, such as urbanisation, greater mobility, and increasing use of the internet and social media may change the profile of the population at higher risk, which emphasises the need for continued monitoring of HIV incidence and sophisticated and evolving assessment of risk, alongside sustained prevention, diagnosis and treatment.

## Methods
### Study design
We used open cohort data from eight independently run general population cohort studies which are members of the Network for Analysing Longitudinal Population-based HIV/AIDS data on Africa (ALPHA): Ifakara and Kisesa in Tanzania, Karonga in Malawi, Kisumu in Kenya, Manicaland in Zimbabwe, Masaka and Rakai in Uganda and uMkhanykaude in South Africa[50]. We used harmonised data from these studies covering people observed between 2005 and 2016, the last date at which most studies collected HIV data.

Ethical approval for this data harmonisation and analysis was granted by the ethics committee of the London School of Hygiene and Tropical Medicine (number 15472) and each study's data collection and analysis was approved by the relevant local committees. Data used in this analysis are not publically available but may be made available to bona fide researchers upon successful application to each study's data access committee.

### Participants
Study participants were included in the incidence cohort if they had participated in the study's HIV testing at least twice whilst resident in the area and their earliest result was HIV negative. In this analysis we used data from people who were aged between 15 and 49.

### Procedures
Several times a year, with the exception of Manicaland[51], each study collected demographic information from household heads in all consenting households within their study areas. For eligible adults, after obtaining informed consent HIV tests and individual surveys, which included questions on HIV-related risk behaviours and socio-demographic characteristics, were carried out at regular intervals, typically between one and three years apart.

Each study is autonomous and data were not standardised prior to collection. Data were harmonised taking into account variation between study sites and over time (see Supplementary Methods for further details).

We identified and constructed a list of common explanatory variables and prioritised those which captured a discrete aspect of behaviour or personal characteristics and were available for most participants. Table 1 shows the measures we reviewed and those we included. We developed a population-level measure to describe the prevalence of untreated HIV infection in people of the opposite sex, resident within the study area,who were in the age range to be sexual partners, constructed using HIV prevalence, ART coverage and the distribution of age gaps between partners. We calculated other population-based measures of risk describing partnership dynamics: the partner acquisition rates[40] and rates for the dissolution of partnerships. For both rates, the denominator was person time in the year before the survey, expressed per 100 person years. The numerators

were, for acquisition, having had sex with a new partner in the last year and, for dissolution, the ending of a partnership (dated to the time of last sex) with a person the respondent had sex with at some point in the year before the survey. We estimated these rates for peers of the same sex (from the same study and calendar period and similar in age) and also for opposite sex potential partners.

## Statistical analysis

The outcome was HIV seroconversion, defined as an HIV positive test in a participant who was resident at the time of the test and who had previously tested negative for HIV. We used multiple imputation to assign a seroconversion date in the interval between the last negative and first positive test dates with a uniform distribution and fitted piecewise exponential regression models to the survival time elapsed between the date of the first negative test and the date of the last negative test or seroconversion[52]. For all estimates we used 70 imputations and combined the estimates using Rubin's rules[53]. Piecewise exponential models were used because they allowed the shape of the hazard function to vary between imputations and captured that variation in the terms for age group.

Subjects had missing data for portions of their follow up time because either information was not collected in every round, participants missed one, or more, survey rounds or due to a small amount of non-response to particular questions. When information was not available, we retained the person time in the analysis and assigned the relevant period of person time to an "unknown" category. People who remained HIV negative were censored at the time of their most recent HIV test.

We wanted to assess if the effects of some risk factors differed by age and sex so analysed four groups of respondents: young men aged 15-24, older men aged 25-49, young women aged 15-24 and older women aged 25-49. Participants whose follow-up time spanned the two age groups contributed to both groups, i.e., they were censored at their 25th birthday in the younger age group and entered the older age group as they turned 25.

We described every risk factor for each age group, study and calendar period using the distribution of person-years for categorical variables and the mean for continuous variables. We estimated the study-specific crude hazard ratio (HR) for new HIV infections and made the same estimates using data pooled from all studies, adjusted for study. We examined the crude HRs for each risk factor and, where study-specific effects appeared to differ based on both HR and p-values, fitted interaction terms to assess effect modification by study.

Risk factors with a consistent pattern of association in the study-specific analyses were included in adjusted models fitted to the data pooled from all studies for each age and sex group. We focussed primarily on the HRs, rather than their associated p-values, since some studies lacked power for some comparisons. Where study-specific HR were different, we included them and captured the heterogeneity of effects between the studies by fitting terms for risk factor-study interactions and retained those with small p-values for the interaction terms. We included a fixed-effect for study because HIV incidence differed between studies. We fitted three piecewise exponential models including age and calendar year: 1) with a fixed effect for study, allowing each study to have a different background incidence rate; 2) with an age and study interaction to relax the assumption of proportional hazards and allow the hazard function to behave differently over age in the different studies and 3) with an age and study interaction plus an interaction between study and prevalence of untreated infection in the opposite sex. We fitted a fourth model which omitted the age and calendar year variables. We hypothesised that age and calendar year are not direct determinants of HIV incidence and by omitting them we aimed to reveal the more proximate determinants of HIV incidence. Lastly, we reviewed the results from these four models for

each age and sex group and judged which was the most parsimonious model- the preferred model- for each group.

Information on partnership dynamics (the numbers and timings of partnerships) was available only for Karonga, Kisesa and Rakai. The analysis was repeated with these studies including the measures of partnership dynamics.

We conducted two sensitivity analyses. To assess the influence each study had on the results from the preferred models we fitted study-specific versions of those models, carried out a meta-analysis on each risk factor estimate and then repeated the meta-analysis using the leave-one-out approach. We also fitted the preferred models to data from 2013-16 only, to see if risk factor associations were different in that period.

Stata version 18.0 was used throughout the analysis.

The funder of the study had no role in study design, data collection, data analysis, data interpretation, or writing of the report.

### Reporting summary

Further information on research design is available in the Nature Portfolio Reporting Summary linked to this article.

## Data availability

A subset of the ALPHA harmonised incidence data used in this study for Karonga, Kisesa, Manicaland and uMkhanyakude have been deposited in the DataFirst database under accession codes mwi-alpha-him-karonga-2002-2017-v1 (Karonga), tza-alpha-himk-1996-2016-v1 (Kisesa), zwe-alpha-himm-1995-2016-v1 (Manicaland), zaf-alpha-himu-2000-2016-v1 (uMkhanyakude) (https://www.datafirst.uct.ac.za/dataportal/index.php/collections/ALPHA). The data are available to bona fide researchers under licensed access via DataFirst. The full set of data collected for this paper are individual participant data that cannot be anonymised whilst maintaining the information needed for this analysis. The study datasets used for this analysis (participant data with identifiers and accompanying data dictionaries) can be requested directly from the individual studies (see https://alpha.lshtm.ac.uk/people/). Requests will be reviewed by each study's data access committee according to their respective criteria for access.

## Code availability

Code used in this paper is available at https://github.com/emmaslay/ALPHA_HIV_Risk_factors_analysis.

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

## Acknowledgements

This work was supported by the Bill and Melinda Gates Foundation [OPP1164897] through salary support for authors (ES, MM, KT, ML, CC).

## Author contributions

ES: Planned and conducted analyses, literature review, harmonised data, wrote first draft of paper, overall responsibility for the paper; CC: Literature review, harmonised data, interpreted results and contributed to drafting the paper; MM: Planned analyses, harmonised data, contributed to data analysis, interpreted the results and contributed to drafting the paper; KR: Harmonised data, triangulated analyses, interpreted results and contributed to drafting the paper; RA: Prepared data for Ifakara, analysed and triangulated results; AD: Prepared data for Karonga, analysed and triangulated results and contributed to drafting the paper; JE: Triangulated analyses, interpreted results and contributed to drafting the paper; CK: Prepared data for Karonga, analysed preliminary results, harmonised data, documented data for sharing; LM: Prepared data for Manicaland, analysed and triangulated results, interpreted results and contributed to drafting the paper; DN: Prepared data for Rakai, analysed and triangulated results, interpreted results and contributed to drafting the paper; DO: Prepared data for Kisumu, analysed and triangulated results and contributed to drafting the paper; AP: Analysed and triangulated results for Karonga, interpreted results and contributed to drafting the paper; GR: Interpreted results and contributed to drafting the paper; MT: Harmonised data, interpreted results and contributed to drafting the paper; KT: Prepared data for Masaka, harmonised pooled data, interpreted results and contributed to drafting the paper; MU: Interpreted results and contributed to drafting the paper; AC: Interpreted results and contributed to drafting the paper; EG: Interpreted results and contributed to drafting the paper; SG: Interpreted results and contributed to drafting the paper; KH: Interpreted results and contributed to drafting the paper; DK: Prepared data for Kisumu, analysed and triangulated results and contributed to drafting the paper; TL: Prepared data for Rakai, analysed and triangulated results, interpreted results and contributed to drafting the paper; RN: Interpreted results and contributed to drafting the paper; JT: Prepared data for Kisesa, analysed and triangulated results, interpreted results and contributed to drafting the paper; JA: Prepared data for Kisumu, analysed and triangulated results; TD: Prepared data for Manicaland, analysed and triangulated results; CK: Prepared data for Kisesa, analysed and triangulated results; CN: interpretation of results for Manicaland; GM: Interpretation of results for Masaka; NMyeza: Prepared data for AHRI; AN: Analysed and triangulated results; GO: Prepared data for Kisumu, analysed preliminary results; AT: Prepared data for Manicaland, analysed preliminary results; CF: Prepared data for Ifakara, analysed and triangulated results; DG Prepared data for AHRI; EMcLean: Prepared data for Karonga, analysed and triangulated results; KB: Interpreted results, triangulated AHRI data; SK: Prepared data for Masaka, analysed preliminary results; NMcGrath: Preliminary analyses for AHRI, interpreted results; DM: Prepared data for Kisesa, analysed preliminary results; AW: Preliminary analyses, interpreted results; EMtuli: Prepared data for Kisesa, analysed preliminary results. All authors had full access to all the data in the study and had final responsibility for the decision to submit for publication. ES, CC, MM, KT have accessed and verified the data.

## Competing interests

SG declares shareholdings in pharmaceutical companies GlaxoSmithKline and Astra Zeneca. The remaining authors declare no competing interests.

## Additional information

[1]Department of Population Health, London School of Hygiene and Tropical Medicine, London, UK. [2]Usher Institute, University of Edinburgh, Edinburgh, UK. [3]Department of Public Health Sciences, Penn State University College of Medicine, Hershey, PA, USA. [4]MRC Centre for Global Infectious Disease Analysis, School of Public Health, Imperial College London, London, UK. [5]Center for Communicable Disease Dynamics, Department of Epidemiology, Harvard T.H. Chan School, Boston, USA. [6]Manicaland Centre for Public Health Research, Biomedical Research and Training Institute, Harare, Zimbabwe. [7]Malawi Epidemiology Intervention Research Unit, Lilongwe, Malawi. [8]Ifakara Health Institute, Ifakara, Tanzania. [9]Rakai Health Sciences Program, Kalisizo, Uganda. [10]Center for Global Health Research (CGHR)-Kisumu, Kenya Medical Research Institute, Kisumu, Kenya. [11]Department of Clinical Research, London School of Hygiene and Tropical Medicine, London, UK. [12]National Institute for Medical Research, Mwanza, Tanzania. [13]Department of Infectious Disease Epidemiology and International Health, London School of Hygiene and Tropical Medicine, London, UK. [14]Africa Health Research Institute, Durban, South Africa. [15]School of Health and Wellbeing, University of Glasgow, Glasgow, UK. [16]DSI-SAMRC & South African Population Research Infrastructure Network (SAPRIN), Durban, South Africa. [17]Kenya Medical Research Institute, Kisumu, Kenya. [18]Department of Health Sciences, University of York, York, UK. [19]Catholic University of Health and Allied Sciences, Mwanza, Tanzania. [24]Deceased: Mark Urassa. ✉e-mail: emma.slaymaker@lshtm.ac.uk

## the ALPHA Network

Georges Reniers[1], Chifundo Kanjala[7], Julie Ambia[20], Tawanda Dadirai[6], Dickman Gareta[14], Coleman Kishamawe[12], Gertrude Mutonyi[21], Njabulo Myeza[14], Constance Nyamukapa[6], Anthony Ndyanabo[9], George Olilo[10], Albert Takaruza[6], Charles Festo[8], Sylvia Kusemererwa[21], Nuala McGrath[14,22,23], Denna Michael[12], Alison Wringe[1] & Emmanuel Mtuli[12]

[20]Bristol Medical School, Faculty of Health Sciences, University of Bristol, Bristol, UK. [21]MRC/UVRI and LSHTM Uganda Research Unit, Entebbe, Uganda. [22]School of Primary Care, Population Sciences and Medical Education & School of Economic, Social and Political Sciences, University of Southampton, Southampton, UK. [23]University of KwaZulu-Natal, Durban, South Africa.

