## [Transparent Peer Review file · Nature Communications]

Individual and population-level risk factors for new HIV infections among adults in Eastern and Southern Africa.

Corresponding Author: Dr Emma Slaymaker

Version 0:

Reviewer comments:

Reviewer #1

(Remarks to the Author)

This paper reports an analysis of risk factors for acquisition of HIV infection using population cohort data from six African countries. Data from eight population cohort studies (under the ALPHA network umbrella) were harmonized – this analysis included data covering the period 2005-2016. The analysis included people aged 15-49 years who had two or more HIV test results in the dataset, the earliest of which was negative. The outcome was HIV seroconversion, and the seroconversion date was assigned to a date between the last negative and first positive test using multiple imputation. A number of individual-level and community-level explanatory variables were selected based on a predefined framework of proximate and distant determinants of HIV acquisition (from literature review).

The analysis included 99,353 individuals (56% female) with total follow-up time 349,201 person years (so mean follow-up around 3.5 years). There were 4612 seroconversions. There was a lot of missing risk factor data, such that sexual behaviour data was only available for approximately 25% of total follow-up time.

A number of factors were associated with HIV acquisition, although not all were consistent across the different sex/age strata. The key findings were consistent with previous data and understanding.

This is an important analysis on a priority public health problem, using the best available data from multiple population cohorts in Africa. The main problem (acknowledged by the authors) is of course how old the data are (ending in 2016), which really limits how these findings can inform public health action now. The other main limitation (again appropriately acknowledged) is the extent of missing risk factor data, which differed by cohort and by sex/age group – this means the findings should be interpreted cautiously.

I have a few comments on the paper that the authors might consider if revising the manuscript.

Major comments

1. Methods, lines 405-410: I couldn't see anywhere in main text or supplementary materials at which geographic level the population-level measures were defined. Were these just developed at the level of the entire population (for each individual cohort) or at some geographic subdivision level? This seems important information to include, and important to bring out in the discussion that, if the former, this might be too broad to capture quite differential risk within each of the communities
2. Results, lines 101-201: In general, I found the results text very dense and quite hard to read, meaning that for the reader the key findings are lost a bit in all the detail. There is too much repetition of results between tables and text. Would strongly recommend pruning this section to highlight only the most noteworthy results.
3. Results, lines 149-201: In general, I found the presentation of results too constrained by the standard rigid statistical framework, such that associations are observed or not observed based on the limits of the 95% CI, but not accounting for different sample sizes and events, and therefore precision of estimates, in the different strata. One example (but not the only one) is the explanation that secondary education was associated with reduced risk for young women (aHR 0.70, 95%CI 0.56-0.87) but no association was observed for young men (aHR 0.72, 95% CI 0.49-1.06). Here the point estimates are very similar but the estimate for young men is less precise given the smaller sample size. So the results are actually compatible with similar associations for young women and young men. In general, I would recommend more nuanced and careful interpretation of the results – in this example of education, I might personally say something like there was strong evidence of an association for young women and weak evidence of an association for young men.

Minor comments

1. Abstract, lines 28-29: Here it is stated that you 'estimated the association between individual and community-level risk factors and HIV incidence...' This suggests the outcome measure (HIV incidence) was a population-level outcome, whereas yours was an individual-level outcome (new HIV infection). Suggest therefore rewording to 'incident HIV infection' or 'HIV acquisition'. This is the first example of this, but this is repeated at various points throughout the paper.
2. Results, lines 102-103: 'Around a quarter of people moved house during the period of observation'. Is this what is referred to as 'residential mobility' further on in this section? If so, would be good to just be clear and consistent with terminology throughout
3. Results, lines 107-108: I don't find it that helpful to lump together the person time with zero and one partner – I would find it more helpful to report specifically the time with zero partner, especially as the number of infections in people with zero partners is quite high (>10% of all new infections were observed in people reporting zero partners in the last year)
4. Results, lines 170-175: I may have got disorientated, but should this paragraph not come under the next subheading ('Adjusted estimates of association with

Reviewer #2

(Remarks to the Author)

In this study, the authors use data from population-based cohorts from 2005-2016 to identify risk factors for incident HIV among adults (15-49 years) in the general population (as opposed to selected, key (or high-risk) sub-populations) from six countries: Kenya, Malawi, South Africa, Tanzania, Uganda and Zimbabwe. The main outcome of interest is HIV incidence, defined as documented seroconversion among cohort members. Overall, the manuscript provides interesting, significant and rigorous findings on both individual- and population-level factors associated with risk of HIV acquisition.

Strengths of the manuscript include use of data available from multiple cohorts across Eastern and Southern Africa, with data for over 349,201 person-years of cohort time, data on sexual partnerships and risk for ~90,000 person-years of cohort time and >4600 seroconversions, relatively clear presentation of the analyses, and sensitivity analyses. Limitations include restriction of analyses to risk factors that were included across the cohorts, and, as a result, inability to include certain factors (such as alcohol use or wealth). In addition, there are several publications evaluating risk factors for seroconversion among a general population in Eastern and Southern Africa from recent, prior universal HIV test and treat trial data that are overlooked and should be cited.

Specific comments/critiques for the authors to address:

- 1) Results section: 49% of the seroconversions came from one cohort study: uMkhanyakude, as presented in Table 1. This should be noted in the "Data available" section of the Results as well.
- 2) Results section: The "prevalence of potential risk factors" section is quite challenging to follow, as presented, with large paragraphs that present data somewhat haphazardly. If the authors could re-organize these two paragraphs either thematically, or by age and sex (similar to how the authors summarize the findings in the Discussion section), in a way that can be followed more easily, this would be helpful (for example, commenting on certain demographics like site of residence, mobility and education first, then moving onto marital status, etc.).
- 3) Results section: Similar to the last comment regarding presentation of results, the sections "Crude estimates of association with HIV incidence in pooled data," and "Adjusted estimates of association with HIV incidence in pooled data," should stick to a parallel structure when presenting risk factors. Otherwise, the results are quite challenging to follow/read.
- 4) Discussion: It would be helpful if the authors could comment further on the potential impact of excluding certain key factors (e.g. wealth, alcohol use, pre-exposure prophylaxis (PrEP) use and sexual practices) in the Discussion section.
- 5) Discussion: On lines 350-351, the authors note, "There are no more recent studies of HIV risk in the general population in sub-Saharan Africa." However, there have been publications from large, population-based universal HIV test and treat trials that evaluated predictors of HIV seroconversion in a general adult population: these should be referenced and included in the Discussion. Examples include
 - a) Nyabuti et al, "Characteristics of HIV seroconverters in the setting of universal test and treat: Results from the SEARCH trial in rural Uganda and Kenya," PLoS One, 2021, PMID: 33544717);
 - b) Skalland et al, "Community- and individual-level correlates of HIV incidence in HPTN 071 (PopART)," Journal of the International AIDS Society, 2023, PMID: 37643290;
 - c) Larmarange et al, "Population-level viremia predicts HIV incidence at the community level across the Universal Testing and Treatment Trials in eastern and southern Africa," Plos Global Public Health, 2023, PMID: 37450436); and
 - d) Cui et al "Predictors of HIV seroconversion in Botswana," AIDS, 2025, PMID: 39497537.
- 6) In the Discussion, the authors appropriately acknowledge in lines 341-342 that, "the risks we have identified may be less important today." However, they then continue as follows in lines 344-347: "Sensitivity analysis limited to 2013-16 data suggests that the risk factors in that period were similar to those identified for the whole period. Many were identified as risks in the late 1990s and early 2000s, at very different phases of the epidemic. Therefore there is nothing to suggest that their importance would have changed since 2016." This last sentence is an assumption that may not be accurate and should be removed or revised. Given that universal antiretroviral therapy for persons with HIV (i.e., regardless of CD4 count) was recommended by WHO in 2015, with implementation of universal ART in many settings starting in 2016, and the role out of

dolutegravir-based therapy (i.e., TLD) began after 2017 in the six countries evaluated – it seems highly plausible that the importance of various risk factors may have changed since 2016. For example, from 2018-onward, in an era of universal, highly potent ART for PWH who know their status, the relative importance of factors associated with not knowing one's diagnosis or an inability to achieve viral suppression despite knowing one's diagnosis (e.g., factors associated with non-retention in care or non-adherence), may be greater than in the pre-2016 period.

Additional minor comments:

Tables:

- Table 1: Please specify that HIV prophylaxis data was “not included” in last column. At present, it simply says, “Little data.”
- Table 2: It would be helpful to show the country associated with each cohort under the “Study name” section.
- Table 4: It would be helpful to bold (or otherwise highlight) the hazard ratios and 95% CIs with significant results.

Reviewer #3

(Remarks to the Author)

We thank the authors for their highly valuable and timely work. By consolidating and harmonizing data across ALPHA member studies and applying a strong analytic framework, they make a significant contribution to population-level estimates of HIV incidence in a high-burden region, particularly in addressing the “lack of geographic representation in the literature” noted in their interesting review.

However, we do have some questions/concerns that we believe the authors should address before the journal proceeds with publication.

INTRODUCTION

In the introduction, the authors introduce community-level risk factors and their effects on individual-level factors: “The prevalence of infectious individuals in the population and the chances of a susceptible person encountering an infectious one will greatly modify the effects of risk behaviours.” However, it is not clear that they intend to study these community-level risk factors (a distinguishing aspect of this work) in the statement of the aim (lines 78–79). We suggest clarifying the statement in line with how it is presented in the abstract.

METHODS

On the handling of missing data, the authors stated the following (Appendix 2, line 163): “We did not impute missing data because we did not believe that would be unbiased.” Since the missing indicator method the authors ultimately used can also introduce bias, especially under circumstances likely applicable in this study (e.g., for covariate missingness due to patient absence/attrition from cohorts, which could conceivably be “missing at random” or “missing completely at random”, particularly if censoring is not informative), could the authors perhaps provide a bit more context on this decision (e.g., why do you believe that missingness, aside from data missing due to questions not being asked in particular rounds of cohort visits, may be “missing not at random” and therefore not addressable using multiple imputation)? Briefly outlining why this approach was considered preferable to multiple imputation, considering the type of missingness encountered, would be helpful. Alternatively, demonstrating the robustness of the main findings via a sensitivity analysis comparing the missing indicator method with multiple imputation could further solidify the conclusions for the reader.

Is the exponential survival model appropriate here (even if allowing departures from a proportional hazards assumption in some models)? Can the authors comment on why they felt this model was appropriately flexible to handle the distribution of survival times in these populations? In addition, the “final” modeling approach used to obtain inference for each age-by-sex group is not entirely clear. For example, the authors describe using multiple approaches for handling age and study effects, and then using the “most parsimonious” model (not clear if this is the model that minimizes AIC or uses some other bias-variance trade-off), but the final form of the models for each age-by-sex group should be noted in Table 2 footnotes (though they can be deduced from Supplementary Tables 8-11). In addition, how were time-varying covariates handled? Participants could seemingly be included in the young and older risk groups, but the numbers of such participants should be reported, including the approach of inclusion (e.g., in the analysis of younger participants, were those who could also be included in the older group censored at the time they aged into the older group?). A bit more clarification would be helpful.

Finally, there may be a significant source of selection introduced because study eligibility hinges on the availability of ≥ 2 HIV tests? Could the authors comment on whether or why they do not believe the receipt of ≥ 2 HIV tests may be related to any additional factors which may be causal parents of HIV incidence? Or, if there are potentially other factors related in this manner (which could lead to a source of selection bias), could the authors describe how they've tried to address this (or why the bias may be negligible)?

RESULTS

There are some potential discrepancies in the numbers reported. For example, Supplementary Figure 2 counts 19,170 pooled men 25–49 years old, while that count is 19,042 in Table 2. Similarly, these counts for women 25–49 years old are reported as 30,995 and 30,702, respectively. In the “Data available” section, the numbers of participants from Ifakara (1,502) and uMkhanyakude (26,778) line up with Supplementary Figure 2 for the former but not the latter (27,101). Moreover, in the “Data available” section, the numbers of men and women are reported at 43,434 and 55,919, respectively. In the abstract, they are reported as 43,135 and 55,354, respectively. If these discrepancies are due to imputed incidence (before presenting stratified/average counts), please clarify; if not, please resolve the discrepancies.

Some descriptive statements under the "Prevalence of potential risk factors" section are not paired with estimates, particularly lines 131–148. I suggest adding estimates consistently throughout to make the descriptions more objective and explicit.

What is the significance of "2013-16s_6 1o" row in Table 3? What does ** denote? Please make sure Tables are readable independent of the text with appropriate footnotes as necessary.

Supplementary Tables 4, 5, 6, 7, 8, 8*, 9, and 10 were unavailable for review. *Two distinct supplementary tables were assigned the same number.

Reviewer #4

(Remarks to the Author)

Version 1:

Reviewer comments:

Reviewer #1

(Remarks to the Author)

The authors have addressed the points raised appropriately and have revised the manuscript accordingly. The manuscript has improved as a result of these changes. Thank you for engaging constructively.

Reviewer #2

(Remarks to the Author)

Overall, the revised manuscript is responsive to comments provided by this reviewer, and the other reviewers. As noted in my prior review, the manuscript provides interesting, significant and rigorous findings on both individual- and population-level factors associated with risk of HIV acquisition.

On review of the revised manuscript, I have two remaining comments:

1. I found this phrase in the Discussion section, line 412-413, confusing and inaccurate:

"There are no studies with recently collected data on HIV risk in the general population in sub-Saharan Africa."

There are several studies evaluating HIV risk in the general population since 2016, including:

a) Recent studies of biomedical HIV prevention within the SEARCH collaboration, such as:

Kakande et al, "A community-based dynamic choice model for HIV prevention improves PrEP and PEP coverage in rural Uganda and Kenya: a cluster randomized trial," JIAS, 2023, PMID: 38054535

b) Koss et al, "Uptake, engagement, and adherence to pre-exposure prophylaxis offered after population HIV testing in rural Kenya and Uganda: 72-week interim analysis of observational data from the SEARCH study" Lancet HIV, 2020, PMID: 32087152 (with data through 2017)

Do the authors mean there are no recently collected data on HIV risk factors and their association with documented HIV seroconversion?

I suggest revising this for clarity and accuracy.

2. Minor revision for clarity: Results section, line 106:

"Primary education accounted for approximately 60% of person-time." I also found this sentence confusing. I assume the author means that 60% of person-time was contributed by people with primary education only? Recommend revising for clarity.

Reviewer #3

(Remarks to the Author)

We appreciate the thorough and thoughtful revisions. The authors have addressed all our prior comments and provided appropriate clarifications and justifications throughout the manuscript and tables. Minor discrepancies in counts appear to have been resolved or explained. Overall, we find the manuscript to be substantially improved and well-prepared for publication. We have no further substantive comments.

Reviewer #4

(Remarks to the Author)

Response to reviewers comments- Nature communications NCOMMS-25-24752

We would like to thank all three reviewers for their very helpful, thoughtful and constructive comments on this paper.

Our responses are in orange below each point.

Reviewer #1 (Remarks to the Author):

This paper reports an analysis of risk factors for acquisition of HIV infection using population cohort data from six African countries. Data from eight population cohort studies (under the ALPHA network umbrella) were harmonized – this analysis included data covering the period 2005-2016. The analysis included people aged 15-49 years who had two or more HIV test results in the dataset, the earliest of which was negative. The outcome was HIV seroconversion, and the seroconversion date was assigned to a date between the last negative and first positive test using multiple imputation. A number of individual-level and community-level explanatory variables were selected based on a predefined framework of proximate and distant determinants of HIV acquisition (from literature review).

The analysis included 99,353 individuals (56% female) with total follow-up time 349,201 person years (so mean follow-up around 3.5 years). There were 4612 seroconversions. There was a lot of missing risk factor data, such that sexual behaviour data was only available for approximately 25% of total follow-up time.

A number of factors were associated with HIV acquisition, although not all were consistent across the different sex/age strata. The key findings were consistent with previous data and understanding.

This is an important analysis on a priority public health problem, using the best available data from multiple population cohorts in Africa. The main problem (acknowledged by the authors) is of course how old the data are (ending in 2016), which really limits how these findings can inform public health action now. The other main limitation (again appropriately acknowledged) is the extent of missing risk factor data, which differed by cohort and by sex/age group – this means the findings should be interpreted cautiously.

I have a few comments on the paper that the authors might consider if revising the manuscript.

Major comments

1. Methods, lines 405-410: I couldn't see anywhere in main text or supplementary materials at which geographic level the population-level measures were defined. Were these just

developed at the level of the entire population (for each individual cohort) or at some geographic subdivision level? This seems important information to include, and important to bring out in the discussion that, if the former, this might be too broad to capture quite differential risk within each of the communities

We appreciate the reviewer for noticing this omission. We have added text to make this clear (line 407). We did the estimation for the whole study population. Whilst there is almost certainly geographical variation in risk across the study areas we do not have good information on how much mixing there is between people who live in different parts of the study area. Without a good way to capture this in our measure we opted for an average across the study area. We have acknowledged this limitation in the discussion (line 332)

2. Results, lines 101-201: In general, I found the results text very dense and quite hard to read, meaning that for the reader the key findings are lost a bit in all the detail. There is too much repetition of results between tables and text. Would strongly recommend pruning this section to highlight only the most noteworthy results.

This is a very fair comment. We structured in this way as it used fewer words but we have now re-written this section and hope the wood now stands out from the trees.

3. Results, lines 149-201: In general, I found the presentation of results too constrained by the standard rigid statistical framework, such that associations are observed or not observed based on the limits of the 95% CI, but not accounting for different sample sizes and events, and therefore precision of estimates, in the different strata. One example (but not the only one) is the explanation that secondary education was associated with reduced risk for young women (aHR 0.70, 95%CI 0.56-0.87) but no association was observed for young men (aHR 0.72, 95% CI 0.49-1.06). Here the point estimates are very similar but the estimate for young men is less precise given the smaller sample size. So the results are actually compatible with similar associations for young women and young men. In general, I would recommend more nuanced and careful interpretation of the results – in this example of education, I might personally say something like there was strong evidence of an association for young women and weak evidence of an association for young men.

Another very fair point. There are so many individual estimates that it is difficult to fully reflect on each while being mindful of the word count so we have rewritten the results to distinguish between results which showed no effect in later models (i.e. HR moved close to 1) and those which still showed an effect but with large p-values.

Minor comments

1. Abstract, lines 28-29: Here it is stated that you 'estimated the association between individual and community-level risk factors and HIV incidence...' This suggests the outcome measure (HIV incidence) was a population-level outcome, whereas yours was an individual-

level outcome (new HIV infection). Suggest therefore rewording to 'incident HIV infection' or 'HIV acquisition'. This is the first example of this, but this is repeated at various points throughout the paper.

We have reworded as suggested.

2. Results, lines 102-103: 'Around a quarter of people moved house during the period of observation'. Is this what is referred to as 'residential mobility' further on in this section? If so, would be good to just be clear and consistent with terminology throughout

We have rephrased the description of this risk factor throughout.

3. Results, lines 107-108: I don't find it that helpful to lump together the person time with zero and one partner – I would find it more helpful to report specifically the time with zero partner, especially as the number of infections in people with zero partners is quite high (>10% of all new infections were observed in people reporting zero partners in the last year)

We have described the two groups separately in the results text, starting on line 108.

4. Results, lines 170-175: I may have got disorientated, but should this paragraph not come under the next subheading ('Adjusted estimates of association with

Yes! We have moved it.

Reviewer #2 (Remarks to the Author):

In this study, the authors use data from population-based cohorts from 2005-2016 to identify risk factors for incident HIV among adults (15-49 years) in the general population (as opposed to selected, key (or high-risk) sub-populations) from six countries: Kenya, Malawi, South Africa, Tanzania, Uganda and Zimbabwe. The main outcome of interest is HIV incidence, defined as documented seroconversion among cohort members. Overall, the manuscript provides interesting, significant and rigorous findings on both individual- and population-level factors associated with risk of HIV acquisition.

Strengths of the manuscript include use of data available from multiple cohorts across Eastern and Southern Africa, with data for over 349,201 person-years of cohort time, data on sexual partnerships and risk for ~90,000 person-years of cohort time and >4600 seroconversions, relatively clear presentation of the analyses, and sensitivity analyses. Limitations include restriction of analyses to risk factors that were included across the cohorts, and, as a result, inability to include certain factors (such as alcohol use or wealth). In addition, there are several publications evaluating risk factors for seroconversion among a general population in Eastern and Southern Africa from recent, prior universal HIV test and treat trial data that are overlooked and should be cited.

Specific comments/critiques for the authors to address:

1) Results section: 49% of the seroconversions came from one cohort study: uMkhanyakude, as presented in Table 1. This should be noted in the "Data available" section of the Results as well.

Done

2) Results section: The "prevalence of potential risk factors" section is quite challenging to follow, as presented, with large paragraphs that present data somewhat haphazardly. If the authors could re-organize these two paragraphs either thematically, or by age and sex (similar to how the authors summarize the findings in the Discussion section), in a way that can be followed more easily, this would be helpful (for example, commenting on certain demographics like site of residence, mobility and education first, then moving onto marital status, etc.).

We agree and, as noted above, we have re-written this section (starting line 100) and structured it more systematically.

3) Results section: Similar to the last comment regarding presentation of results, the sections "Crude estimates of association with HIV incidence in pooled data," and "Adjusted estimates of association with HIV incidence in pooled data," should stick to a parallel structure when presenting risk factors. Otherwise, the results are quite challenging to

follow/read.

As above, we have restructured this section.

4) Discussion: It would be helpful if the authors could comment further on the potential impact of excluding certain key factors (e.g. wealth, alcohol use, pre-exposure prophylaxis (PrEP) use and sexual practices) in the Discussion section.

Done, we have added a paragraph in the discussion starting on line 379

5) Discussion: On lines 350-351, the authors note, "There are no more recent studies of HIV risk in the general population in sub-Saharan Africa." However, there have been publications from large, population-based universal HIV test and treat trials that evaluated predictors of HIV seroconversion in a general adult population: these should be referenced and included in the Discussion. Examples include

a) Nyabuti et al, "Characteristics of HIV seroconverters in the setting of universal test and treat: Results from the SEARCH trial in rural Uganda and Kenya," PLoS One, 2021, PMID: 33544717);

b) Skalland et al, "Community - and individual - level correlates of HIV incidence in HPTN 071 (PopART)," Journal of the International AIDS Society, 2023, PMID: 37643290;

c) Larmarange et al, "Population-level viremia predicts HIV incidence at the community level across the Universal Testing and Treatment Trials in eastern and southern Africa," Plos Global Public Health, 2023, PMID: 37450436); and

d) Cui et al "Predictors of HIV seroconversion in Botswana," AIDS, 2025, PMID: 39497537.

We have rephrased this statement – "There are no studies with recently collected data on HIV risk in the general population in sub-Saharan Africa." and have incorporated the more recent papers into the discussion.

6) In the Discussion, the authors appropriately acknowledge in lines 341-342 that, "the risks we have identified may be less important today." However, they then continue as follows in lines 344-347: "Sensitivity analysis limited to 2013-16 data suggests that the risk factors in that period were similar to those identified for the whole period. Many were identified as risks in the late 1990s and early 2000s, at very different phases of the epidemic. Therefore there is nothing to suggest that their importance would have changed since 2016." This last sentence is an assumption that may not be accurate and should be removed or revised. Given that universal antiretroviral therapy for persons with HIV (i.e., regardless of CD4 count) was recommended by WHO in 2015, with implementation of universal ART in many settings starting in 2016, and the role out of dolutegravir-based therapy (i.e., TLD) began

after 2017 in the six countries evaluated – it seems highly plausible that the importance of various risk factors may have changed since 2016. For example, from 2018-onward, in an era of universal, highly potent ART for PWH who know their status, the relative importance of factors associated with not knowing one’s diagnosis or an inability to achieve viral suppression despite knowing one’s diagnosis (e.g., factors associated with non-retention in care or non-adherence), may be greater than in the pre-2016 period.

We agree, of course, that things may always change and treatment options may well have altered the risk factors. In fact that was one of the motivations for this analysis. However since most of the changes subsequent to universal treatment are still contingent on diagnosis, and these populations already had high levels of diagnosis prior to 2016, we do believe the factors we identified are likely to remain important. We have rephrased this as “This consistency over time could mean that they remain important going forwards.” (line 408)

Additional minor comments:

Tables:

- Table 1: Please specify that HIV prophylaxis data was “not included” in last column. At present, it simply says, “Little data.”
- Table 2: It would be helpful to show the country associated with each cohort under the “Study name” section.
- Table 4: It would be helpful to bold (or otherwise highlight) the hazard ratios and 95%CIs with significant results.

All done

Reviewer #3 (Remarks to the Author):

We thank the authors for their highly valuable and timely work. By consolidating and harmonizing data across ALPHA member studies and applying a strong analytic framework, they make a significant contribution to population-level estimates of HIV incidence in a high-burden region, particularly in addressing the "lack of geographic representation in the literature" noted in their interesting review.

However, we do have some questions/concerns that we believe the authors should address before the journal proceeds with publication.

INTRODUCTION

In the introduction, the authors introduce community-level risk factors and their effects on individual-level factors: "The prevalence of infectious individuals in the population and the chances of a susceptible person encountering an infectious one will greatly modify the effects of risk behaviours." However, it is not clear that they intend to study these community-level risk factors (a distinguishing aspect of this work) in the statement of the aim (lines 78–79). We suggest clarifying the statement in line with how it is presented in the abstract.

Line 78 added "individual and community-level"

METHODS

On the handling of missing data, the authors stated the following (Appendix 2, line 163): "We did not impute missing data because we did not believe that would be unbiased." Since the missing indicator method the authors ultimately used can also introduce bias, especially under circumstances likely applicable in this study (e.g., for covariate missingness due to patient absence/attrition from cohorts, which could conceivably be "missing at random" or "missing completely at random", particularly if censoring is not informative), could the authors perhaps provide a bit more context on this decision (e.g., why do you believe that missingness, aside from data missing due to questions not being asked in particular rounds of cohort visits, may be "missing not at random" and therefore not addressable using multiple imputation)? Briefly outlining why this approach was considered preferable to multiple imputation, considering the type of missingness encountered, would be helpful. Alternatively, demonstrating the robustness of the main findings via a sensitivity analysis comparing the missing indicator method with multiple imputation could further solidify the conclusions for the reader.

The missing data we encountered arises for two reasons, questions not being fielded in a round (or in an entire study) or because the reference period for a question is shorter than the interval between survey rounds for an individual. There is also a small amount of item non-response, which we would assume is biased.

As Reviewer 3 notes, gaps in data collection where questions were not asked are not random with respect to time and study. The bias may be larger where we have missing data

because the reference period for the question doesn't cover the time elapsed since the respondent was last asked the question. Where the gap arises because the interval between surveys is longer than the reference period (as happens in, for example, Kisesa where questions are typically asked for the last 12 months but the surveys are at least two years apart) this is less likely to cause an important bias because everyone has the same amount of missing data. It is still correlated with study, and time, because the interval between the surveys in some studies has changed over time. However, perhaps the bigger issue, is that it is very common for people to miss survey rounds, even among the most committed participants. Informal information on the reasons for missing rounds leads us to understand it is due to family and work responsibilities, travel and health concerns. Since these may well be associated with HIV risk, both in terms of altered exposure and behaviour, we do not feel confident to assume they are missing at random. We have added some text to this effect in Appendix 2 (halfway down page 11).

Is the exponential survival model appropriate here (even if allowing departures from a proportional hazards assumption in some models)? Can the authors comment on why they felt this model was appropriately flexible to handle the distribution of survival times in these populations?

The shape of the hazard functions varies considerably between men and women, between studies, over time and, crucially, between imputations, so using a model with a shape parameter was not a good option as it did not fit on all imputations. Likewise splines-based approaches encounter the same problem across imputations. (We have some extensive methodological work demonstrating this, which we plan to publish this year.) The piecewise exponential was chosen because it did not require us to make an a priori assumption about the shape of the hazard function but allowed us to describe it from the data using age and calendar time and provided a method for capturing that variation in the shape of the hazard function across the imputations. Additionally it is a simple and well understood method, which we felt would aid communication of the results to a non-statistically minded reader. We did not use a Cox model as we expected to find non-proportional hazards, though in the end this was only encountered in one model. We have added a sentence to the methods to explain this (line 485).

In addition, the "final" modeling approach used to obtain inference for each age-by-sex group is not entirely clear. For example, the authors describe using multiple approaches for handling age and study effects, and then using the "most parsimonious" model (not clear if this is the model that minimizes AIC or uses some other bias-variance trade-off), but the final form of the models for each age-by-sex group should be noted in Table 2 footnotes (though they can be deduced from Supplementary Tables 8-11).

We have now explained this more clearly at the start of the results section on line 188. We did not use AIC or another formal way to select our model because this is not possible with the imputation approach that we used.

In addition, how were time-varying covariates handled?

Everything except sex was time-varying, and we have now stated this in Table 1. In terms of how we allocated the episodic information from the surveys to the longitudinal follow-up time, I have now indicated this in Table 1 and added some information in Appendix 3: Supplementary Methods.

Participants could seemingly be included in the young and older risk groups, but the numbers of such participants should be reported, including the approach of inclusion (e.g., in the analysis of younger participants, were those who could also be included in the older group censored at the time they aged into the older group?). A bit more clarification would be helpful.

It is correct that anyone who was observed either side of their 25th birthday features in the analysis for both age groups and yes, respondents were accordingly left- or right-censored at their 25th birthday. Text to this effect has been added at line 496.

Finally, there may be a significant source of selection introduced because study eligibility hinges on the availability of ≥ 2 HIV tests? Could the authors comment on whether or why they do not believe the receipt of ≥ 2 HIV tests may be related to any additional factors which may be causal parents of HIV incidence? Or, if there are potentially other factors related in this manner (which could lead to a source of selection bias), could the authors describe how they've tried to address this (or why the bias may be negligible)?

We agree with the reviewer that participation bias could be a problem (all residents are eligible so there is no selection). To address this we estimated the study- and time-specific probability of returning for a second test, by age and sex, among people who had had one test. We assessed how these probabilities varied over time and by characteristics of the participants and, although there were some differences, we found no systematic patterns. Very few people participated in the survey but then refused HIV testing, so we have little information on behaviour for people who were not tested. This means it is not possible to look at whether characteristics differ between those who can be included in the incidence cohort and those who are excluded. To incorporate some information on participation in this analysis we included the age-, sex-, study- and time-specific probability of retesting in the regression models. It did not have an effect in the adjusted models and mention of this step fell victim to the word count in this paper. We could reinstate this, it is described more fully in another paper under preparation.

RESULTS

There are some potential discrepancies in the numbers reported. For example, Supplementary Figure 2 counts 19,170 pooled men 25–49 years old, while that count is 19,042 in Table 2. Similarly, these counts for women 25–49 years old are reported as 30,995 and 30,702, respectively. In the "Data available" section, the numbers of participants from Ifakara (1,502) and uMkhanyakude (26,778) line up with Supplementary Figure 2 for the former but not the latter (27,101). Moreover, in the "Data available" section, the numbers

of men and women are reported at 43,434 and 55,919, respectively. In the abstract, they are reported as 43,135 and 55,354, respectively. If these discrepancies are due to imputed incidence (before presenting stratified/average counts), please clarify; if not, please resolve the discrepancies.

The differences between the numbers in Sup Fig 2 (the flowcharts) and Table 2 are indeed because the flowcharts describe the numbers of respondents in each category whereas Table 2 contains the means across imputations. We have now added that to the caption for Table 2 to make that clear. The difference between the data available section and the abstract and the numbers of uMkhanyakude was an error that crept in when updating results which we have now resolved.

Some descriptive statements under the "Prevalence of potential risk factors" section are not paired with estimates, particularly lines 131–148. I suggest adding estimates consistently throughout to make the descriptions more objective and explicit.

We have redrafted this section (starting line 100) and have included estimates more consistently

What is the significance of "2013-16s_6 1o" row in Table 3? What does ** denote? Please make sure Tables are readable independent of the text with appropriate footnotes as necessary.

This was a redundant Stata reference to a variable that was omitted from the model and it is now deleted

Supplementary Tables 4, 5, 6, 7, 8, 8*, 9, and 10 were unavailable for review. *Two distinct supplementary tables were assigned the same number.

These were the tables in the excel file and we are sorry this was not available for review. There was only one Table 8 and we have corrected the numbering in the list of captions in the Appendix.

Reviewer #4 (Remarks to the Author):

Reviewers 1 and 3 had no comments for us and, in response to comments from reviewer 2, we have changed the sentence in the discussion (line 412-413) to read "There are no studies with recently collected data on HIV risk factors and their association with documented HIV seroconversion in the general population in sub-Saharan Africa.". In the Results section (line 106) we have made the suggested clarification about the percentage of person time with primary education.